# Interplay between *ATRX* and *IDH1* mutations governs innate immune responses in diffuse gliomas

Seethalakshmi Hariharan[1,2,9], Benjamin T. Whitfield [3,9], Christopher J. Pirozzi[1,4], Matthew S. Waitkus [1,2], Michael C. Brown[1,2], Michelle L. Bowie[1,2], David M. Irvin[3], Kristen Roso[1,4], Rebecca Fuller[1,2], Janell Hostettler[1,2], Sharvari Dharmaiah[3], Emiley A. Gibson [1,2], Aaron Briley[1,2], Avani Mangoli [1,2], Casey Fraley[1,2], Mariah Shobande[1,2], Kevin Stevenson[1,5], Gao Zhang [1,2], Prit Benny Malgulwar [3], Hannah Roberts[3], Martin Roskoski[1,2], Ivan Spasojevic [6,7], Stephen T. Keir[1,2], Yiping He[1,4], Maria G. Castro[8], Jason T. Huse[3] ✉ & David M. Ashley [1,2] ✉

Stimulating the innate immune system has been explored as a therapeutic option for the treatment of gliomas. Inactivating mutations in *ATRX*, defining molecular alterations in *IDH*-mutant astrocytomas, have been implicated in dysfunctional immune signaling. However, little is known about the interplay between ATRX loss and *IDH* mutation on innate immunity. To explore this, we generated ATRX-deficient glioma models in the presence and absence of the *IDH1^{R132H}* mutation. ATRX-deficient glioma cells are sensitive to dsRNA-based innate immune agonism and exhibit impaired lethality and increased T-cell infiltration in vivo. However, the presence of *IDH1^{R132H}* dampens baseline expression of key innate immune genes and cytokines in a manner restored by genetic and pharmacological IDH1^{R132H} inhibition. IDH1^{R132H} co-expression does not interfere with the ATRX deficiency-mediated sensitivity to dsRNA. Thus, ATRX loss primes cells for recognition of dsRNA, while IDH1^{R132H} reversibly masks this priming. This work reveals innate immunity as a therapeutic vulnerability of astrocytomas.

Adult-type diffuse gliomas are a diverse group of tumors that account for more than 80% of primary CNS malignancies[1]. They are subclassified based on key molecular alterations, namely isocitrate dehydrogenase (*IDH*) mutation and codeletion of the 1p and 19q chromosomal arms (1p/19q codeletion)[2], falling into 3 primary groups: 1) *IDH*-wildtype glioblastoma, 2) *IDH*-mutant, 1p/19q co-deleted oligodendroglioma, and 3) *IDH*-mutant, 1p/19q non-codeleted astrocytoma[3,4]. *IDH*-wildtype glioblastoma is composed primarily of high-grade, clinically aggressive tumors, while both *IDH*-mutant disease groups tend to exhibit lower-grade histo-pathological features at diagnosis. Treatment for all glioma variants involves some combination of surgery, radiation, and alkylating

[1]The Preston Robert Tisch Brain Tumor Center, Duke University Medical Center, Durham, NC, USA. [2]Department of Neurosurgery, Duke University Medical Center, Durham, NC, USA. [3]Departments of Pathology and Translational Molecular Pathology, University of Texas MD Anderson Cancer Center, Houston, TX, USA. [4]Department of Pathology, Duke University Medical Center, Durham, NC, USA. [5]Molecular Physiology Institute, Duke University Medical Center, Durham, NC, USA. [6]PK/PD Core Laboratory, Duke Cancer Institute, Duke University Medical Center, Durham, NC, USA. [7]Department of Medicine – Oncology, Duke University Medical Center, Durham, NC, USA. [8]Department of Neurosurgery, University of Michigan Medical Center, Ann Arbor, MI, USA. [9]These authors contributed equally: Seethalakshmi Hariharan, Benjamin T. Whitfield. ✉e-mail: jhuse@mdanderson.org; david.ashley@duke.edu

chemotherapy[5,6]. However, adult gliomas invariably recur, at which point overall prognosis is poor.

Immunotherapy holds considerable potential in the context of central nervous system (CNS) malignancies[7]. Traditional immunotherapies, such as checkpoint blockade, rely on reviving adaptive immune constituents like T-cells, and have seen great success in otherwise difficult-to-treat tumors[8]. However, these adaptive immune therapies have failed to extend survival in gliomas[9]. Despite these disappointments, other unique immunotherapy modalities have shown early promise in the CNS, such as oncolytic viral therapy[10–12]. While oncolytic viruses can selectively lyse tumor cells, their therapeutic potency may be due to their ability to activate the innate immune system through pattern recognition receptors (PRRs)[13,14]. These receptors detect pathogen associated molecular patterns (PAMPs) and drive multipotent interferon responses, activating multiple arms of the immune system. Prominent examples of PRRs include stimulator of interferon genes (STING), Rig-I-like receptors (RLR), and Toll-like receptors (TLR), with multiple forms of PRR agonism being tested clinically[15–18].

Inactivation of the SWI/SNF chromatin remodeler gene, α-thalassemia retardation X-linked (ATRX), represents a common glioma-associated molecular alteration with the potential to substantially impact the tumor microenvironment[19,20]. ATRX is mutated in more than 80% of IDH-mutant astrocytomas, a large portion of pediatric high-grade gliomas (HGG) and a subset of IDH-wildtype glioblastomas[19]. Recent studies have demonstrated that loss of epigenetic regulators in general can potentiate responses to immunotherapy[21,22]. Of note, loss of ATRX appears to alter immune infiltration, cytokine secretion, and chromatin availability to key immune gene regions[23,24]. Interestingly, IDH mutations, which almost invariably arise with ATRX mutations in adult gliomas, have been shown to suppress leukocyte chemotaxis and infiltration[25]. While the association between mutations in ATRX and IDH1 has been known for over a decade, the details of their interaction and basis for frequent co-occurrence in astrocytomas remains unclear. Furthermore, the implication of these mutations on immune-therapeutic responsiveness is understudied.

In this investigation, we leverage multiple glioma models to demonstrate that ATRX deficiency leads to increased innate immune signaling and cytokine secretion in response to dsRNA-based innate immune agonism. This finding correlates with prolonged survival, suggesting a connection between ATRX deficiency, innate immune signaling, and survival. Furthermore, we show that IDH1 mutation can mask these pro-inflammatory effects, and that inhibition of mutant IDH1 relieves this immune suppression. Taken together, these findings reveal the therapeutic potential of targeting the innate immune system for a large, molecularly defined glioma subclass.

## Results

### Astrocytomas are immunologically inflamed relative to oligodendrogliomas

Prior single cell profiling has shown that IDH and ATRX-mut astrocytomas exhibit increased macrophage and microglial infiltration compared to IDH-mut, ATRX-WT oligodendrogliomas[26]. To determine if ATRX mutations associate with inflammatory signaling pathways in glioma patients, we first performed gene set enrichment analysis (GSEA) on TCGA bulk RNAseq data from IDH1-mutant low-grade gliomas (LGG). The majority of the ATRX-mut astrocytomas exhibited enrichment of gene ontology (GO) terms associated with immune-related pathways, including activation of innate immune response, PRR activation, and response to IFNα and β (Fig. 1a; Supplementary Fig. 1a, b, Supplementary Data 1). By contrast, ATRX-WT/ 1p-19q codeleted oligodendrogliomas exhibited weaker correlations with these immune-related transcriptional signatures. With the notable exception of loci included in the oligodendroglioma-defining 1p/19q codeletion

event (IRF3, JAK1, ISG15, IFNL1, and IFNL2), we found a general paucity of mutations, high-level amplifications, and/or deep deletions associated with ATRX mutations (Supplementary Fig. 2), indicating that the effects of glioma-associated molecular alterations like IDH mutation and ATRX deficiency are likely not mediated via genetic mutations or copy number variations.

To determine the relative contributions of tumor (glioma) vs immune cells in driving the differential inflammatory signatures noted in Fig. 1a, we queried single cell RNAseq data[26,27] from IDH-mutant oligodendrogliomas and astrocytomas. Relative to oligodendrogliomas, in IDH-mutant astrocytomas, immune cell clusters were enriched for gene sets associated with inflammatory response, IFNα response, IFNγ response and TNFα signaling via NFκB; whereas tumor cell clusters exhibited minimal enrichment for immune related molecular networks (Fig.1b; Supplementary Data 2). This lack of enrichment for immune related gene sets may be a result of expression of immunosuppressive IDH mutations in astrocytic tumors, while the tumor microenvironment is still immunologically active. Overall, these data echo earlier findings[26], pointing to a higher degree of immune cell infiltration in IDH-mut, ATRX-mut astrocytomas relative to IDH-mut, ATRX-WT oligodendrogliomas. These findings may indicate a role for ATRX deficiency in promoting an immune-reactive phenotype.

### Generation of ATRX-depleted glioma models

To investigate the role of ATRX deficiency in regulating inflammation in gliomas, we developed both mouse and human experimental systems using glioma cell lines that exhibit intact innate immune signaling pathways. Using a CRISPR/Cas9 approach, we generated single cell Atrx-KO clones—KO-A and KO-B—isolated from the murine CT2A glioma cell line and a polyclonal ATRX-KO derivative of the human M059J glioma cell line, along with corresponding controls. Additionally, we developed a genetically engineered murine glioma model, leveraging RCAS/Ntv-a retroviral transduction to express the oncogene platelet-derived growth factor A (PDGFA), an shRNA targeting TpS3, and Cre-recombinase in nestin-positive cells within the brains of isogenic mice harboring either intact or floxed Atrx loci[28]. This strategy yielded ATRX-intact and deficient gliomas that could then be subjected to ex vivo culture as well as serial transplantation. All ATRX-deficient murine and human cell line reagents demonstrated robust depletion of ATRX protein (Fig. 1c, d). Interestingly, ATRX deficiency modestly impaired in vitro growth (Fig. 1e, f).

### ATRX loss impairs glioma growth in vivo in a manner largely dependent on the immune microenvironment

Next, we evaluated the extent to which ATRX inactivation impacts glioma growth in vivo. Abundant genetic evidence supports the classification of ATRX as a tumor suppressor in human cancer[29]. However, the majority of adult gliomas harboring ATRX deficiency exhibit relatively indolent biology at initial diagnosis despite inexorable malignant progression over time[30]. Consistent with this behavior, we found that Atrx knockout was associated with extended survival in immunocompetent mice subjected to orthotopic allografting with CT2A cell derivatives (Fig. 2a; Supplementary Fig. 3a). Moreover, mice harboring ATRX-deficient gliomas in the context of our RCAS/Ntv-a genetically engineered model also exhibited extended survival relative to ATRX-intact counterparts, in both de novo and re-injection contexts (Fig. 2b; Supplementary Fig. 3b, c). Loss of ATRX expression was validated in both CT2A and RCAS/ Ntv-a end-stage tumors (Supplementary Fig. 3d, e). In both cases, Atrx KO glioma cells demonstrated depleted nuclear ATRX labeling by immunohistochemistry, with retained expression in non-neoplastic cellular constituents. By contrast, ATRX-intact control tumors exhibited uniformly strong ATRX expression in tumor cell nuclei.

To probe the molecular and cellular mechanisms underlying impaired in vivo growth in the ATRX-deficient context, we performed RNA-seq on explanted tumors from RCAS/Ntv-a model and cultured

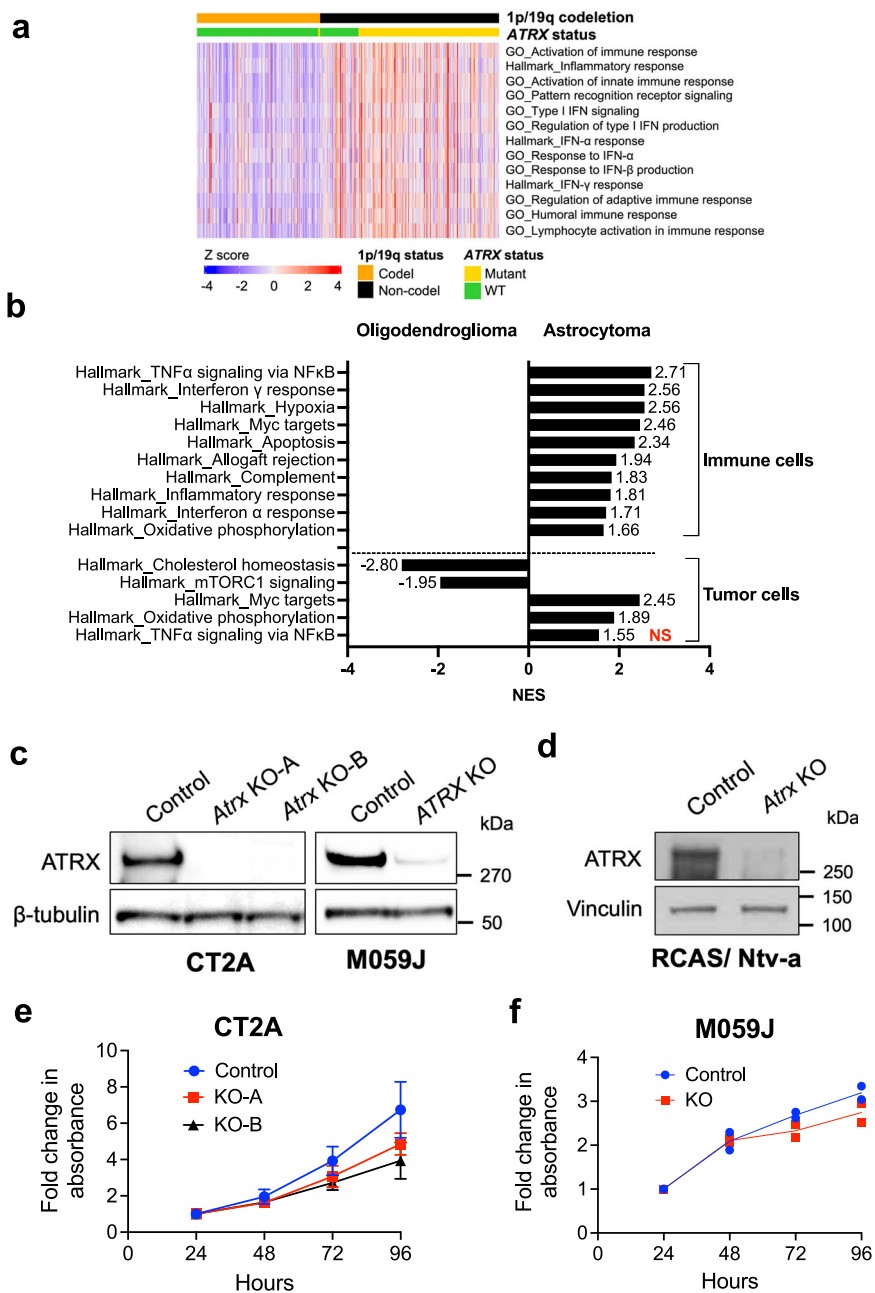

**Fig. 1 | Astrocytomas are immunologically engaged compared to oligoden-drogliomas. a** Heatmap from ssGSEA showing enrichment for various innate immune related gene sets from CNS/Brain TCGA LGG PanCancer Atlas Study, comparing 1p-19q noncodel/ *IDH* mutant/ *ATRX* mutant astrocytomas (*n* = 191) to 1p-19q codel/ *IDH* mutant/ *ATRX* WT oligodendrogliomas (*n* = 164). A relatively smaller number of 1p-19q codel/ *IDH* mutant/ *ATRX* mutant (*n* = 3) and 1p-19q noncodel/ *IDH* mutant/ *ATRX* WT (*n* = 52) are also included in the analysis for comparison. About 55% (28 out of 52) of 1p-19q noncodel/ *IDH* mutant/ *ATRX* WT astrocytomas and 62% (118 out of 191) of 1p-19q noncodel/ *IDH* mutant/ *ATRX* mutant astrocytomas exhibited positive enrichment for these pathways, as shown in Supplementary Data 1. **b** GSEA Hallmark gene sets of scRNAseq data from immune cell clusters and tumor cell clusters from *IDH* mutant oligodendroglioma (GSE70630; *n* = 6) and *IDH* mutant astrocytoma (GSE89567; *n* = 10) patient tumor samples. Normalized enrichment scores (NES) for gene sets that have adjusted *p* value < 0.05 (except tumor cells - Hallmark_TNFa signaling via NFkB highlighted in red) are shown. *P*-value estimation is based on an adaptive multi-level split Monte-

Carlo scheme, while Benjamini-Hochberg procedure is used to correct for multiple hypothesis testing. Results of GSEA analysis are included in Supplementary Data 2. **c** Representative Western blots using lysates from ATRX-intact and ATRX-depleted mouse CT2A cells and human M059J cells showing total ATRX expression. KO-A and KO-B represent two ATRX-deficient CT2A single cell clones used in this study. β-tubulin serves as the loading control. *N* = 3 independent experiments. **d** Representative Western blots using lysates from RCAS/Ntv-a cell lines derived from tumor-bearing *Atrx*^+/+^ Cre mice (Control) and *Atrx*^fl/fl^ + Cre mouse (*Atrx* KO) showing total ATRX expression. Vinculin serves as the loading control. *N* = 3 independent experiments. **e**, **f** Growth assays of CT2A CRISPR control (Ctrl) or *Atrx*-KO cells (KO-A, KO-B) (**e**) and M059J CRISPR control (Ctrl) and *ATRX*-KO (KO) cells (**f**) in culture for 96 h. Data indicates fold change values normalized to corresponding 24h absorbance value for each cell line. Mean ± SEM values are shown for CT2A, and individual data points are shown for M059J; *n* = 3 independent experiments for CT2A and *n* = 2 independent experiments for M059J. Source data are provided as a Source Data file.

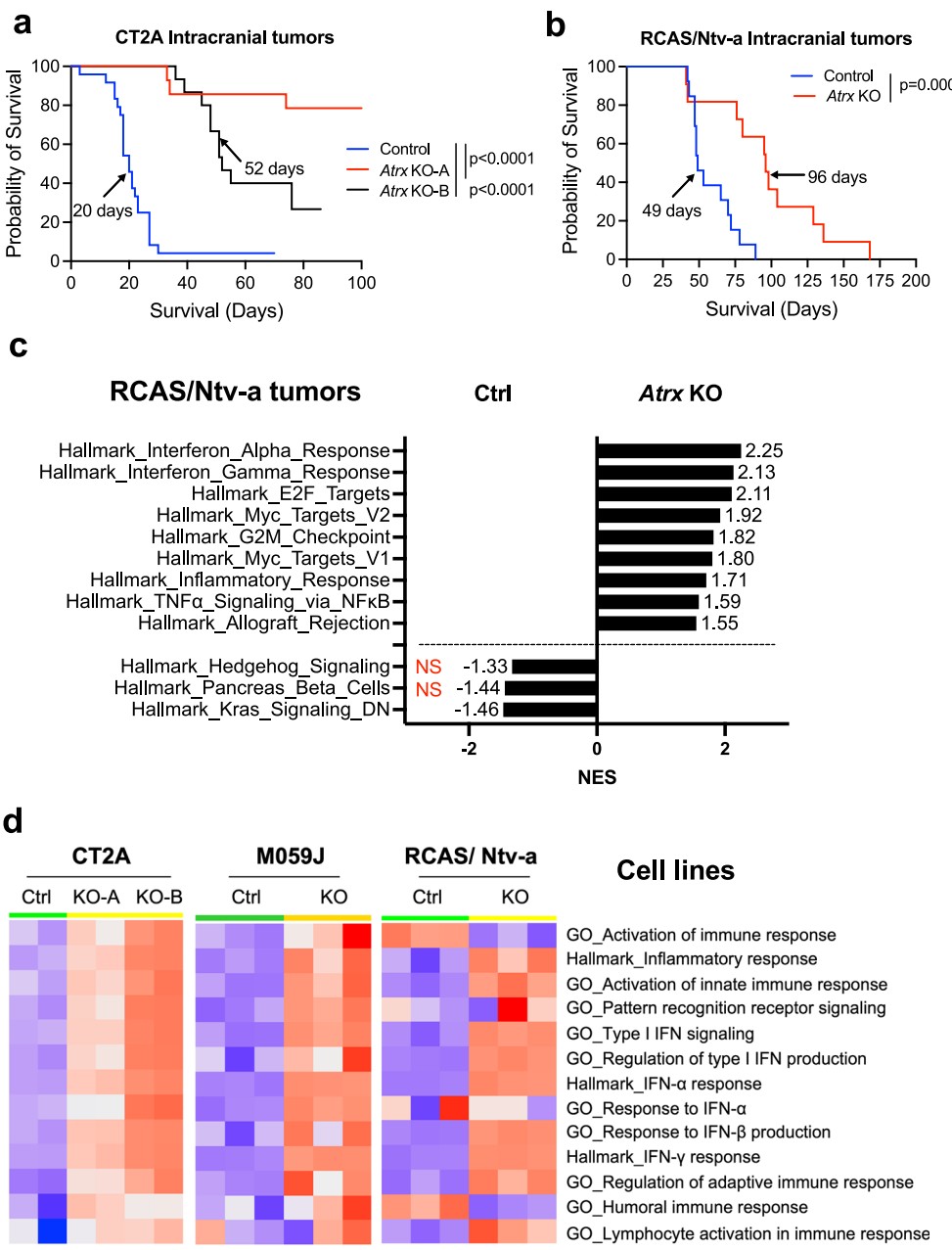

**Fig. 2 | ATRX depletion is associated with improved survival in vivo.**
**a** Kaplan–Meir survival curves for C57BL/6 mice bearing intracranial CT2A CRISPR control (Control) (*n* = 24), *Atrx*-KO clones, KO-A (*n* = 14) and KO-B (*n* = 15) tumors. Median survival is indicated in days for each group except for the *Atrx* KO-A group. *P*-values representing group comparisons were calculated using log-rank test. **b** Kaplan–Meir survival curves for C57BL/6 mice bearing intracranial RCAS/ Ntv-a *Atrx*+/+ (Ctrl) (*n* = 13) and *Atrx*fl/fl (KO) (*n* = 11) tumors. Median survival is indicated in days for both the groups. *P*-values representing group comparisons were calculated using log-rank test. **c** GSEA of Hallmark gene sets from RCAS/ Ntv-a *Atrx*-KO tumors (*n* = 3) compared to *Atrx*WT (Ctrl) tumors (*n* = 3). Normalized enrichment scores (NES) for gene sets that have adjusted *p* value < 0.05 (except

Hallmark_Hedgehog signaling and Hallmark_Pancreas Beta cells, highlighted in red) are shown. *P*-value estimation is based on an adaptive multi-level split Monte-Carlo scheme, while Benjamini-Hochberg procedure is used to correct for multiple hypothesis testing. Results of GSEA analysis are included in Supplementary Data 3. **d** Heatmap from ssGSEA showing enrichment for various innate immune related gene sets in CT2A *Atrx* KO-A and KO-B clones, M059J *ATRX*-KO cells and RCAS/Ntv-a *Atrx*-KO cells compared to their respective *ATRX*WT counterparts. *n* = 2 technical replicates for CT2A expression data; *n* = 3 technical replicates for M059J expression data; *n* = 3 consecutive cell passages for RCAS/Ntv-a cell line expression data. Source data are provided as a Source Data file.

cell lines from CT2A, M059J and RCAS model systems. GSEA of RCAS/ Ntv-a tumors demonstrated differential engagement of cell cycle pathways in ATRX-deficient models relative to ATRX-intact counterparts, implicating tumor cell autonomous molecular mechanisms (Fig. 2c; Supplementary Data 3). However, we also found increased GSEA correlations with a variety of immune signaling networks in

ATRX-deficient models, including the GO terms, inflammatory signaling, PRR signaling, activation of innate immune response, type I interferon signaling and response to IFNα and β (Fig. 2d; Supplementary Fig. 4a, b). Analysis of differential expression and baseline protein levels of genes involved in innate immune signaling indicated increased expression of RIG-I, MDA-5 (gene name - *Ifih1*), STAT1 and

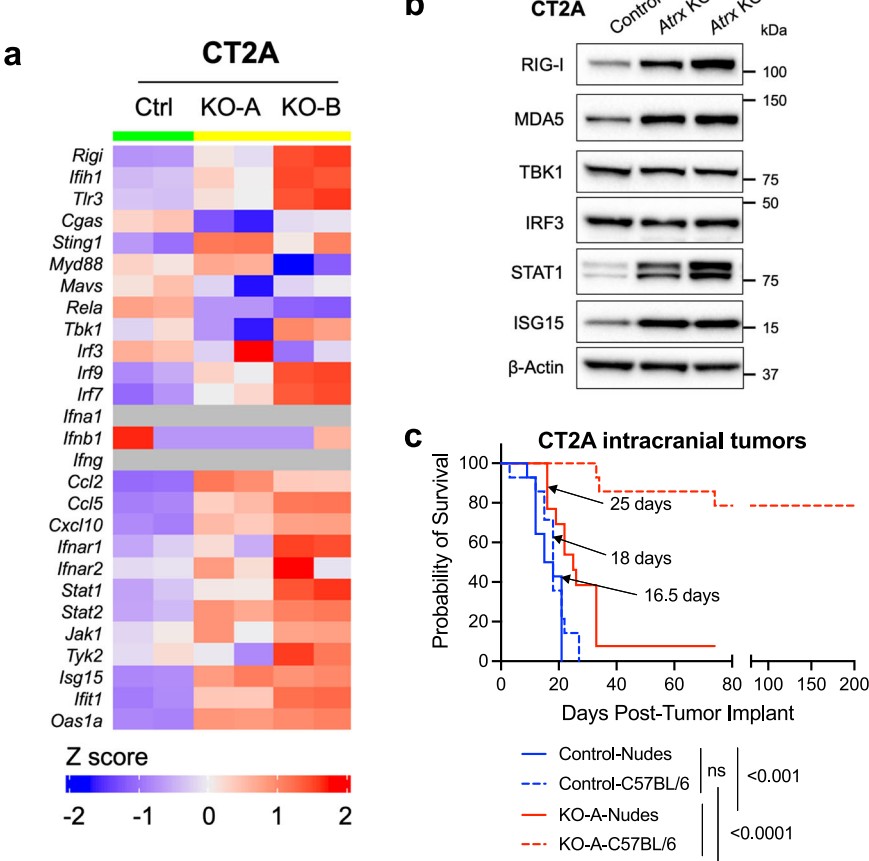

**Fig. 3 | ATRX loss is associated with increased baseline inflammation in vitro.** **a** Heatmap showing differential expression of immune-related genes in CT2A CRISPR control (Ctrl), *Atrx* KO-A and *Atrx* KO-B cells. *n* = 2 technical replicates. **b** Representative Western blots using lysates from CT2A CRISPR ctrl (Ctrl) and *Atrx*-KO clones, KO-A and KO-B screened for proteins involved in innate immune signaling. β-actin serves as the loading control. *N* = 3 independent experiments. Densitometry values are indicated in Supplementary Fig 5d. **c** Kaplan–Meir survival curves for nude and C57BL/6 mice bearing intracranial CT2A CRISPR ctrl (Control) or *Atrx* KO-A tumors. *N* values: Control-Nudes = 14, Control-C57BL/6 = 14, KO-A-Nudes = 13, KO-A-C57BL/6 = 14. Median survival is indicated in days for all the groups except the *Atrx* KO-A-C57BL/6 group. *P*-values represent group comparisons calculated using log-rank test. Control-Nudes vs Control-C57BL/6 – *p* = 0.3324 (ns not significant); *Atrx* KO-A-Nudes vs *Atrx* KO-A-C57BL/6 - *p* < 0.0001; Control-Nudes vs *Atrx* KO-A-Nudes – *p* = 0.0009; Control-C57BL/6 vs *Atrx* KO-A-C57BL/6 - *p* < 0.0001. Source data are provided as a Source Data file.

ISG15 in both CT2A and M059J *Atrx*/*ATRX*-KO cells compared to *Atrx*/*ATRX*$^{WT}$ counterparts, while increased expression of *Ifih1* and *Stat1* was observed in RCAS/Ntv-a *Atrx*-KO cells (Fig. 3a, b; Supplementary Fig. 5a–e). Moreover, several other genes involved in immune regulation, including *Jak1, Irf7, Irf9* and chemokines like *Ccl2, Ccl5* and *Cxcl10* were induced in CT2A *Atrx*-KO, M059J *ATRX*-KO, and RCAS/Ntv-a *Atrx*-KO cells (Fig. 3a; Supplementary Fig. 5a, b).

To delineate the extent to which tumor cell-autonomous and immune microenvironmental factors impaired in vivo growth of our ATRX-deficient glioma models, we compared survival among mice carrying *Atrx*$^{WT}$ or *Atrx* KO-A tumors in either the nude background, which lacks cellular immunity, or the C57BL/6 immune-intact background for our CT2A model. We found that differences in allograft growth between ATRX-intact and ATRX-deficient glioma were much less apparent in nude hosts compared to C57BL/6 immunocompetent hosts (Fig. 3c). Taken together, these findings strongly suggest that the indolent growth of ATRX-deficient glioma models is largely attributable to immune microenvironmental effects.

**ATRX loss leads to upregulation of cytokine secretion, increased immune cell infiltration and mitigation of glioma growth by cellular immune effectors cells**

Having implicated the immune microenvironment as a potential causative factor promoting indolence in ATRX-deficient glioma, we sought to determine the extent to which ATRX depletion promotes increased immune cell infiltration in our in vivo models. To this end, we used flow cytometry to analyze dissociated *Atrx*$^{WT}$ and *Atrx*-KO CT2A tumor-bearing brain hemispheres from immunocompetent mice 14 days after intracranial implantation (Fig. 4a; Supplementary Fig. 6). At this time point, gliomas were not macroscopically evident in *Atrx*-KO CT2A tumor-bearing mice, prompting our analysis of the entire xenografted hemisphere for both *Atrx*-KO and *Atrx*$^{WT}$ cases. ATRX loss led to a significant increase in CD3 + T cells and, in particular, CD4 + T-cells within tumor-bearing hemispheres in allografted mice, while macrophage infiltration was reduced. Trends toward increases in multiple T-cells subsets were also observed in RCAS tumor xenografts by the same experimental paradigm, although differences did not reach formal statistical significance (Supplementary Fig. 8a). By contrast, flow cytometry analysis demonstrated reduced overall macrophage levels in ATRX-deficient xenografts relative to ATRX-intact controls (Fig. 4a). We reasoned that this discrepancy might simply reflect the lower tumor size of ATRX-deficient xenografts, and indeed by immunohistochemistry, we found higher levels of CD45 and F4/80 cell labeling in both *Atrx* KO-A and KO-B tumors than in CRISPR controls (Fig. 4b; Supplementary Fig. 7a, b). Once again, these findings were supported by similar trends in the RCAS model, and CD3 IHC results approached statistical significance (*p* = 0.0649) (Fig. 4b, c; Supplementary Fig. 8b). To assess the extent to which increased T-cells

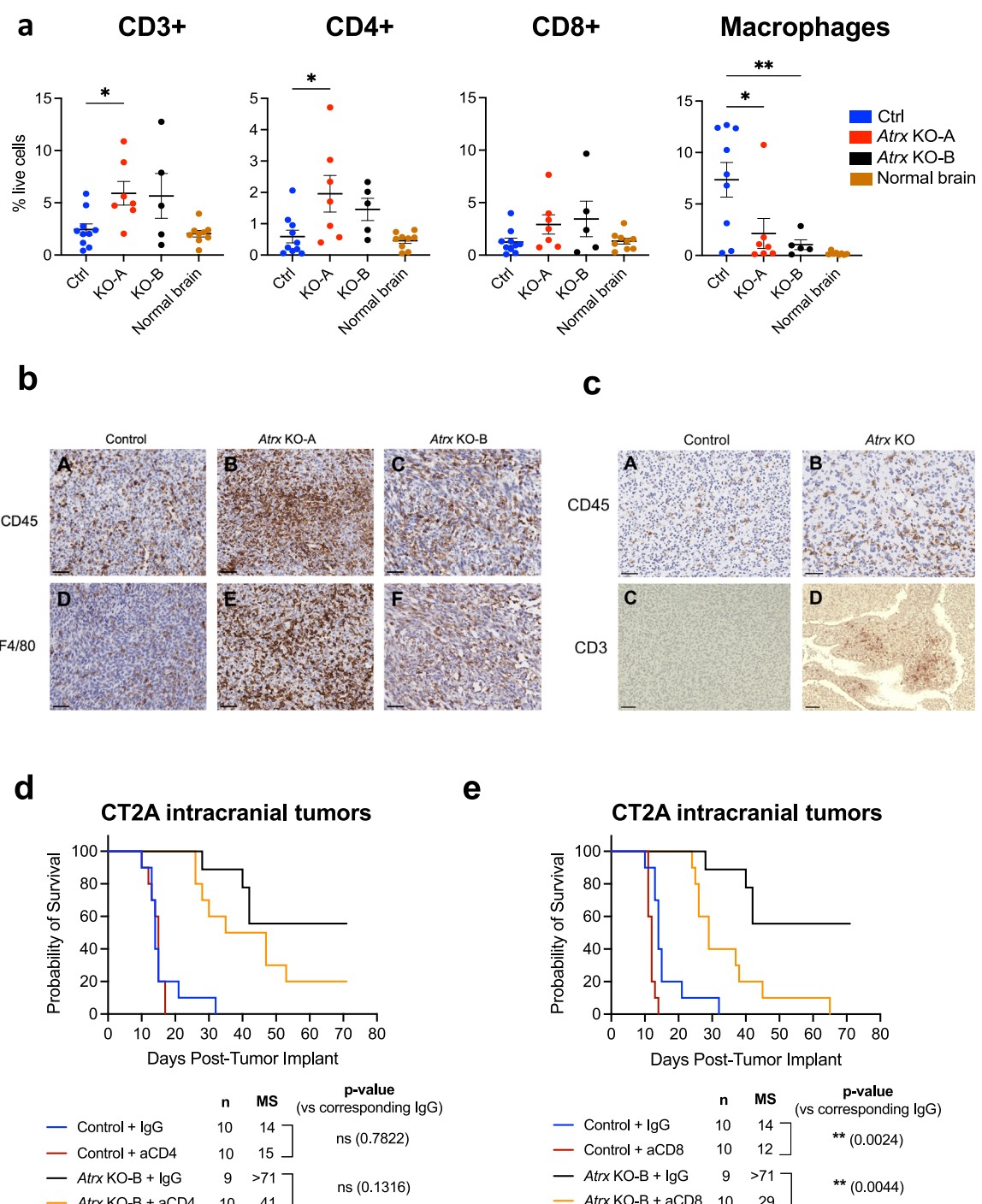

**Fig. 4 | ATRX deficiency leads to increased immune cell infiltration. a** Flow cytometry analysis of CT2A CRISPR control, *Atrx* KO-A or KO-B tumor-bearing hemispheres harvested 14 days post-intracranial implant showing percent live cell density of CD3 + , CD4+ and CD8 + T-cells and macrophages (CD45 + , CD3/19neg, NK1.1neg, CD11bhi, CD45hi). Gating strategy is provided in Supplementary Fig. 6. *n* = 10 for CRISPR control – CD3 + , CD4 + , CD8+ markers; *n* = 9 for CRISPR control - macrophage markers; *n* = 7 for *Atrx* KO-A - all markers; *n* = 5 for *Atrx* KO-B – all markers; *n* = 9 for normal brain - CD3 + , CD4 + , CD8+ markers; *n* = 8 for normal brain – macrophage markers. Data are presented as mean ± SEM. Asterisks denote significant one-way ANOVA with Dunnett's post-hoc test comparing CRISPR control with *Atrx* KO tumor-bearing hemispheres (\**p* < 0.05, \*\**p* < 0.01). CD3 + : Control vs *Atrx* KO-A – *p* = 0.0296; CD4 + : Control vs *Atrx* KO-A – *p* = 0.0117; Macrophages: Control vs *Atrx* KO-A – *p* = 0.0167, Control vs *Atrx* KO-B – *p* = 0.0086.

**b** Representative IHC images for CD45 (A, B, C) and F4/80 (D, E, F) expression (brown) in CT2A CRISPR Control (*Atrx*^WT) (A, D), *Atrx* KO-A (B, E) and KO-B (C, F) tumors, with hematoxylin counter-staining (blue). Scale bar: 50μm. Number of brains subjected to IHC per group: CRISPR Control – *n* = 3; *Atrx* KO-A – *n* = 2; *Atrx* KO-B – *n* = 3. **c** Representative IHC images for CD45 (A, B) and CD3 (C, D) expression (brown) in RCAS/ Ntv-a control (A, C) and *Atrx*-KO (B, D), with hematoxylin counter-staining. Scale bar: 50μm. Number of brains subjected to IHC per group: RCAS/ Ntv-a control – *n* = 6; *Atrx* KO – *n* = 4. **d, e** Kaplan–Meir survival curves for C57BL/6 mice bearing intracranial CT2A CRISPR ctrl (Control) or *Atrx* KO-B tumors treated with 250 µg of isotype control IgG, anti-CD4 antibody (**d**) or anti-CD8 antibody (**e**) as per schema in Supplementary Fig. 9. N Number of mice per group; MS median survival in days. *P*-values represent group comparisons calculated using log-rank test. Source data are provided as a Source Data file.

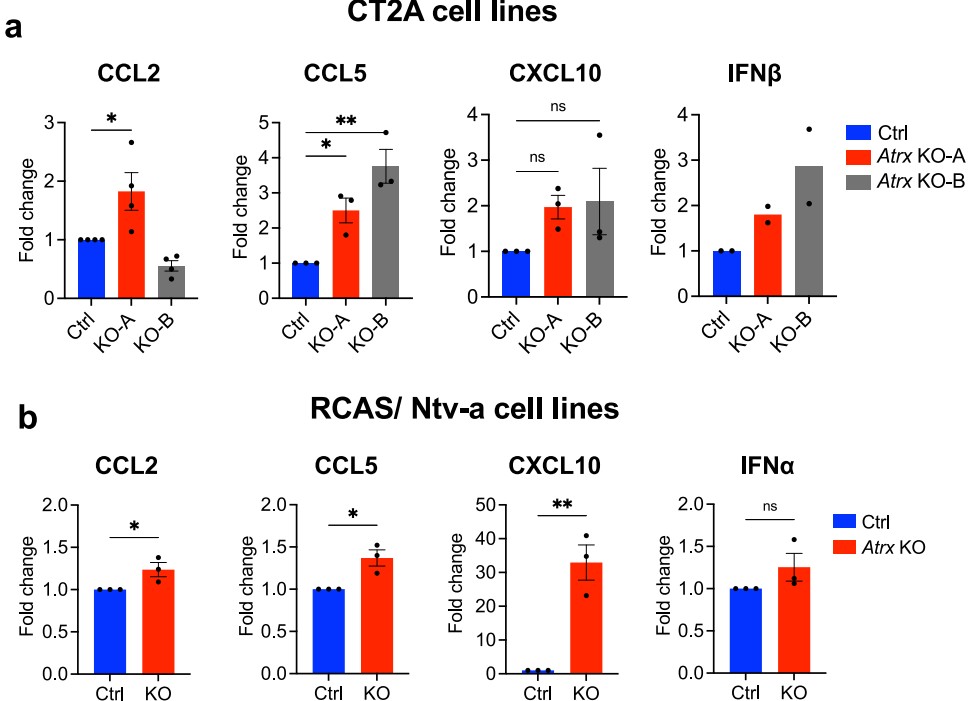

**Fig. 5 | ATRX deficiency leads to increased cytokine secretion in isogenic cell lines. a, b** Cytokine levels in conditioned media from untreated CT2A CRISPR control (Ctrl), *Atrx* KO-A and KO-B clones (**a**), and RCAS/Ntv-a *Atrx*+/+ (Ctrl) and *Atrx*-/- (KO) cell lines (**b**). Supernatant cytokines were analyzed by cytokine bead arrays for antiviral and proinflammatory cytokines. Data indicates fold change values normalized to CRISPR control (Ctrl; CT2A model) or *Atrx*+/+ (Ctrl; RCAS/Ntv-a), shown as mean ± SEM. *N* values: Independent experiments for CT2A model: *n* = 4 for CCL2; *n* = 3 for CCL5, CXCL10; *n* = 2 for IFNβ; Technical replicates for RCAS/Ntv-

a model: *n* = 3 for all cytokines. Asterisks denote significant one-way ANOVA with Dunnett's post-hoc test (CT2A model) and significant unpaired, two-tailed *t*-test (RCAS/Ntv-a model) (*$p < 0.05$; **$p < 0.01$). **a** CCL2: CRISPR Control vs *Atrx* KO-A - $p = 0.0255$; CCL5: CRISPR Control vs *Atrx* KO-A - $p = 0.0378$, CRISPR Control vs *Atrx* KO-B - $p = 0.0023$. **b** CCL2: Control vs *Atrx* KO - $p = 0.0475$; CCL5: Control vs *Atrx* KO - $p = 0.0185$; CXCL10: Control vs *Atrx* KO - $p = 0.0036$. Source data are provided as a Source Data file.

in ATRX-deficient tumor-bearing mice promoted overall survival, we depleted CD4+ and CD8 + T-cells in C57BL/6 immunocompetent hosts bearing CT2A *Atrx*WT or *Atrx* KO-B tumors (Fig. 4d, e; Supplementary Fig. 9). CD4 + T-cell depletion led to a modest survival reduction in *Atrx* KO-B-bearing mice that was not statistically significant. More strikingly, CD8 + T-cell depletion resulted in significantly reduced survival for both *Atrx*WT and *Atrx* KO-B tumor-bearing mice. These findings further implicate cellular immune effectors in mitigating glioma growth, particularly in the ATRX-deficient context.

Moreover, RNA-seq analysis across CT2A, M059J, and RCAS/Ntv-a cell lines revealed upregulated transcripts for multiple chemokines/ cytokines, including CCL2, CCL5, CXCL10 and IFNβ in the ATRX-deficient context, pointing to an underlying mechanism for immune cell recruitment (Fig. 3a; Supplementary Fig. 5a). We then confirmed upregulated cytokine secretion in ATRX-deficient isogenics for CT2A and RCAS/Ntv-a cell lines (Fig. 5a, b). Taken together, these findings mirror those of human gliomas (Fig. 1a), where ATRX deficiency was associated with immune cell infiltration and pro-inflammatory signaling.

## ATRX depletion leads to enhanced innate responses to dsRNA agonists

As described above, *ATRX*-KO cell lines demonstrated increased expression of RIG-I and MDA5 dsRNA sensors. However, the extent to which ATRX loss influences dsRNA-triggered innate immune responses, such as pro-inflammatory cytokine secretion, has not been explored. Using commercially available murine and human panels, we found that treatment with either low molecular weight (LMW) or high molecular weight (HMW) poly(I:C)—a synthetic dsRNA analog— dramatically upregulated secretion of cytokines including CCL2, CCL5 and CXCL10 in CT2A *Atrx*-KO clones (Fig. 6a; Supplementary Fig. 10a),

IFNβ, IL28α/β, IL29 and IL6 in M059J *ATRX*-KO cells (Fig. 6b), and CXCL1, CCL2 and CXCL10 in RCAS/Ntv-a *Atrx*-KO cells (Fig. 6c). Poly(I:C) treatment also induced higher STAT1 phosphorylation and ISG15 expression, both markers of innate immune signaling, in all three ATRX-deficient isogenic models. Finally, poly(I:C) strongly induced phosphorylation of IRF3, an innate immune transcriptional regulator phosphorylated early during antiviral signaling, in *Atrx/ATRX*-KO CT2A and M059J cells, further indicating that these effects were due to enhanced sensing of dsRNA. (Fig. 6d–f; Supplementary Fig. 10b). Taken together, these observations reveal that ATRX inactivation sensitizes glioma cells to dsRNA.

## Co-expression of IDH1R132H with ATRX loss attenuates baseline innate immune signaling in gliomas

Recent work has shown that mutant *IDH1* cooperates with ATRX loss in promoting the alternative lengthening of telomere (ALT) phenotype in gliomas, highlighting mechanisms of pathogenic interplay between the two molecular abnormalities[31]. Moreover, mutant *IDH1* has been associated with an immunosuppressive phenotype characterized by decreased tumor infiltration by T-cells and reduced expression of cytotoxic T lymphocyte-associated genes, pro-inflammatory cyto-kines, and chemokines[25,32]. These phenotypes, along with many others, have been linked to production of the oncometabolite D-2-hydroxyglutarate (D2HG) by mutant IDH1[33,34]. To investigate the impact of co-occurring *IDH1* and *ATRX* mutations on innate immune signaling in glioma, we generated CT2A cells expressing mutant IDH1R132H in both *Atrx*WT and *Atrx*-KO genetic backgrounds. IDH1R132H expression and D2HG production were confirmed in relevant cell lines by immunoblotting and mass spectrometry, respectively (Fig. 7a, b). Expression of IDH1R132H did not alter in vitro proliferation, regardless of *ATRX* status (Fig. 7c).

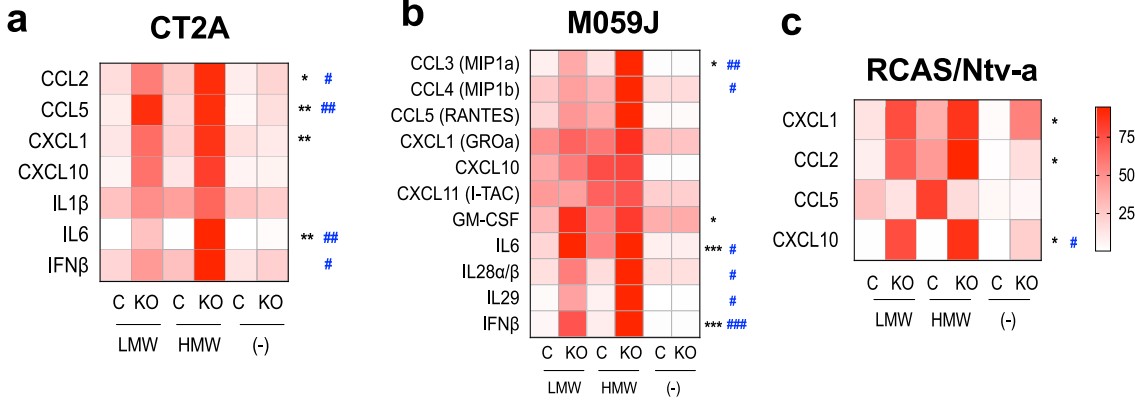

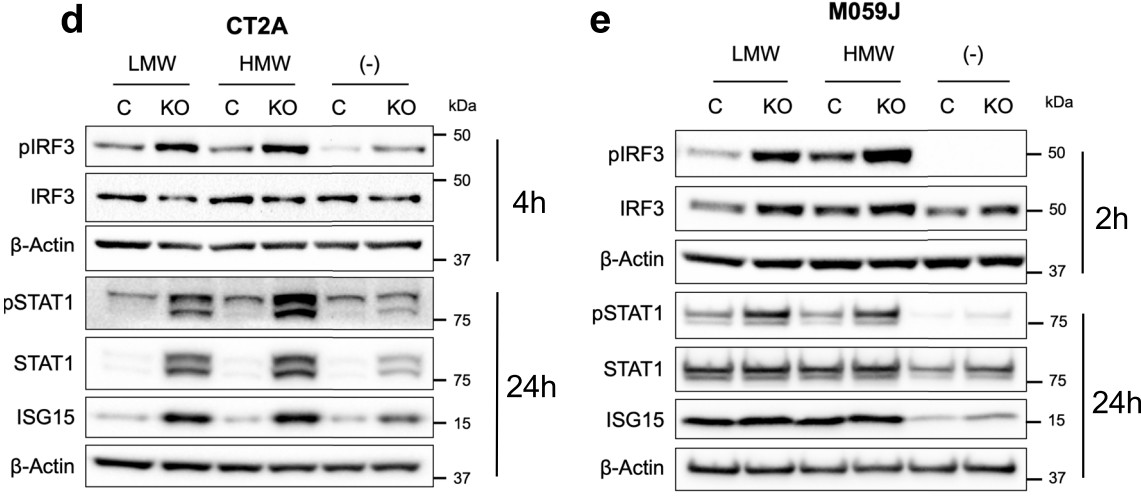

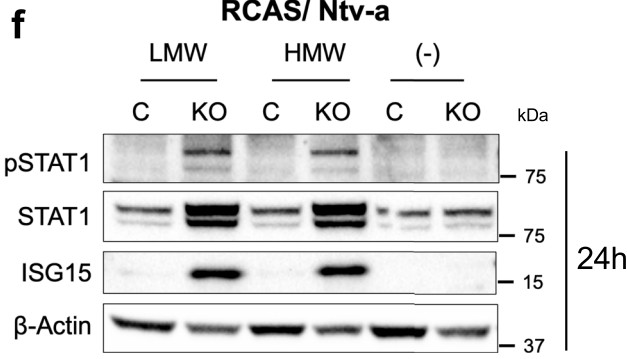

**Fig. 6 | ATRX depletion sensitizes cells to poly(I:C), a dsRNA agonist. (a, b, c)** Cytokine levels in conditioned media from CT2A CRISPR control and *Atrx* KO-A cells (**a**), M059J CRISPR control and *ATRX*-KO cells (**b**) and RCAS/Ntv-a *Atrx*⁺/⁺ or *Atrx*⁻/⁻ cell lines (**c**) treated with 10 µg/ml poly(I:C) LMW or HMW for 24 h. Supernatant cytokines were analyzed by cytokine bead arrays for antiviral and proinflammatory cytokines. CT2A: *n* = 4 independent experiments for CCL2 and IL6, *n* = 3 independent experiments for all other cytokines; M059J and RCAS/Ntv-a: *n* = 3 independent experiments. For heatmap generation, maximum values for each cytokine were set to 100%. Asterisks & hashtags denote significant one-way ANOVA with Sidak's post-hoc test comparing both poly(I:C) LMW (*) and poly(I:C) HMW (#) between *ATRX*-KO and control (*ATRX*^WT) cell lines (*, #*p* < 0.05; **, ##*p* < 0.01; ***, ###*p* < 0.0001). Only cytokines and signaling proteins with observed induction after treatment with poly(I:C) are included. **d, e, f** Representative Western blots using lysates from CT2A CRISPR control and *Atrx* KO-A cells (d), M059J CRISPR control and *ATRX*-KO cells (e) and RCAS/Ntv-a *Atrx*⁺/⁺ or *Atrx*⁻/⁻ cells (f) treated with poly(I:C)

LMW or HMW for 2 h (M059J), 4 h (CT2A) or 24 h CT2A, M059J, RCAS/Ntv-a), screened for pIRF3/IRF3, pSTAT1/STAT1 and ISG15 involved in innate immune signaling. β-actin serves as the loading control. *N* = 3 independent experiments for CT2A, M059J and RCAS/Ntv-a models. **a** CT2A - *p*-values for CRISPR Control + poly(I:C) LMW vs *Atrx* KO-A + poly(I:C) LMW: CCL2 – 0.0196, CCL5 – 0.0065, CXCL1 – 0.0081, IL6 – 0.018; CRISPR Control + poly(I:C) HMW vs *Atrx* KO-A + poly (I:C) HMW: CCL2 – 0.0452, CCL5 – 0.0034, IL6 – 0.0192, IFNβ– 0.0448. **b** M059J – *p*-values for CRISPR Control + poly(I:C) LMW vs *ATRX* KO + poly (I:C) LMW: CCL3 – 0.0111, GM-CSF – 0.0515, IL6 – 0.0002, IFNβ– 0.0008; CRISPR Control + poly(I:C) HMW vs *ATRX* KO + poly(I:C) HMW: CCL3 – 0.0037, CCL4 – 0.0442, IL6 – 0.0166, IL28α/β– 0.0106, IL29 – 0.0379, IFNβ– 0.0002. **c** RCAS/Ntv-a - *p*-values for *Atrx*⁺/⁺ + poly(I:C) LMW vs *Atrx*⁻/⁻ + poly(I:C) LMW: CXCL1 – 0.0122, CCL2 – 0.0477, CXCL10 – 0.0124; *Atrx*⁺/⁺ + poly(I:C) HMW vs *Atrx*⁻/⁻ + poly(I:C) HMW: CXCL10 – 0.0127. Source data are provided as a Source Data file.

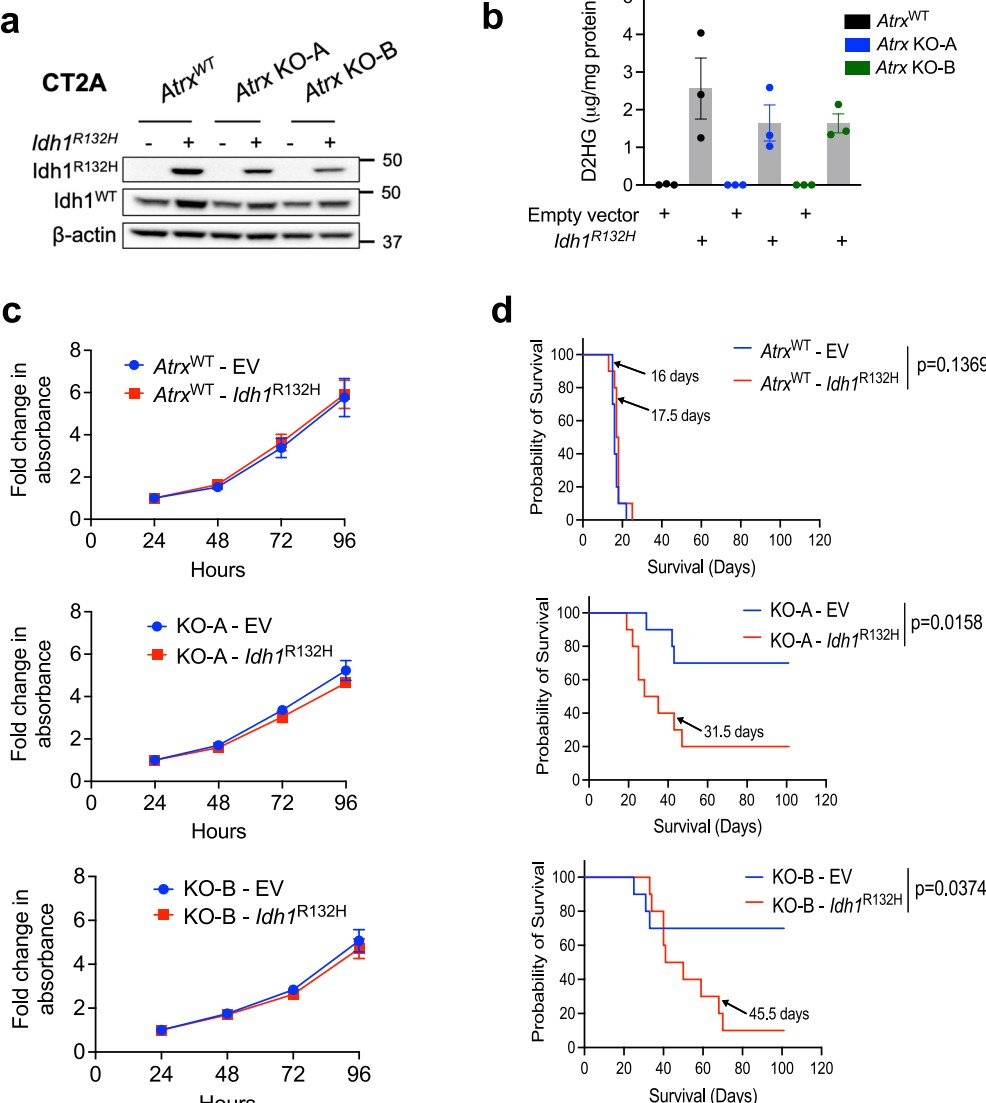

**Fig. 7 | Generation & characterization of IDH1$^{R132H}$-expressing CT2A cells.**
**a** Lysates from CT2A CRISPR control (*Atrx*$^{WT}$) or *Atrx*-KO cells (KO-A, KO-B) expressing exogenous IDH1$^{R132H}$, or empty vector were screened by immunoblotting for IDH1$^{R132H}$ and wildtype IDH1 expression. β-actin serves as the loading control. Representative blots are shown; $n = 3$ independent experiments. **b** D2HG levels in cell pellets from CT2A CRISPR control (*Atrx*$^{WT}$) or *Atrx*-KO cells (KO-A, KO-B) expressing IDH1$^{R132H}$ or empty vector in culture for 72 h, normalized to total protein. $n = 3$ independent experiments. Data are presented as mean ± SEM. **c** Growth assays of CT2A CRISPR control (*Atrx*$^{WT}$) or *Atrx*-KO cells (KO-A, KO-B) expressing IDH1$^{R132H}$

or empty vector in culture for 96 hrs. Data represents mean ± SEM. $n = 3$ independent experiments. **d** Kaplan–Meir survival curves for C57BL/6 mice bearing intracranial CT2A CRISPR ctrl-empty vector control (*Atrx*$^{WT}$ - EV), CRISPR ctrl-*Idh1*$^{R132H}$ (*Atrx*$^{WT}$ - *Idh1*$^{R132H}$), *Atrx* KO-A - empty vector control (KO-A - EV), *Atrx* KO-A - *Idh1*$^{R132H}$ (KO-A - *Idh1*$^{R132H}$), *Atrx* KO-B - empty vector control (KO-B - EV) and *Atrx* KO-B - *Idh1*$^{R132H}$ (KO-B - *Idh1*$^{R132H}$) tumors. n = 10 for each group. *P*-values represent group comparisons calculated using the log-rank test. Median survival is indicated in days for all the groups except the KO-A - *Idh1*$^{R132H}$ and KO-B - *Idh1*$^{R132H}$ groups. Source data are provided as a Source Data file.

To determine the effect of these combined molecular alterations on in vivo tumor growth, *Atrx*$^{WT}$ and *Atrx*-KO CT2A cell lines, with or without IDH1$^{R132H}$, were intracranially allografted into immunocompetent mice. Resulting tumors exhibited largely identical histopathology, regardless of their underlying genetics (Supplementary fig. 11). However, we found that co-expression of IDH1$^{R132H}$ shortened overall survival only in the *Atrx* KO context (Fig. 7d), implying a selective role in glioma progression in ATRX-deficient gliomas.

RNAseq analysis of our isogenic CT2A lines indicated that while ATRX deficiency upregulated various innate and adaptive immune-related gene sets, co-expression of IDH1$^{R132H}$ mitigated these effects (Fig. 8a; Supplementary Fig. 12, 13). For instance, differentially expressed genes involved in innate immune signaling like *Ddx58, Ifih1, Tlr3, Myd88, Irf7, Stat1, Isg15*, enriched in *Atrx*-

KO cells, were downregulated in *Atrx*-KO/ *Idh1*$^{R132H}$ cells (Fig. 8b; Supplementary Fig. 14a). IDH1$^{R132H}$ expression was also associated with decreased baseline levels of key innate immune pathway proteins like RIG-I, MDA5, STAT1 and ISG15 in both *Atrx*$^{WT}$ and *Atrx*-KO contexts, although expression of IRF3 was unaffected across cell lines (KO-A - Fig. 8c, Supplementary Fig. 14c; KO-B - Supplementary Fig. 14b, d). Moreover, secretion of cytokines like CCL2, CCL5, CXCL10 and IFNβ was similarly downregulated upon IDH1$^{R132H}$ expression (KO-A - Fig. 8d; KO-B - Supplementary Fig. 14e). These findings indicate that co-existent IDH1$^{R132H}$ tempers ATRX-deficient inflammatory signaling in gliomas, further supporting the notion that mutant *IDH1* confers an immunosuppressive phenotype and pointing to a pathogenically relevant immunological interplay between the defining molecular alterations of *IDH*-mutant astrocytomas.

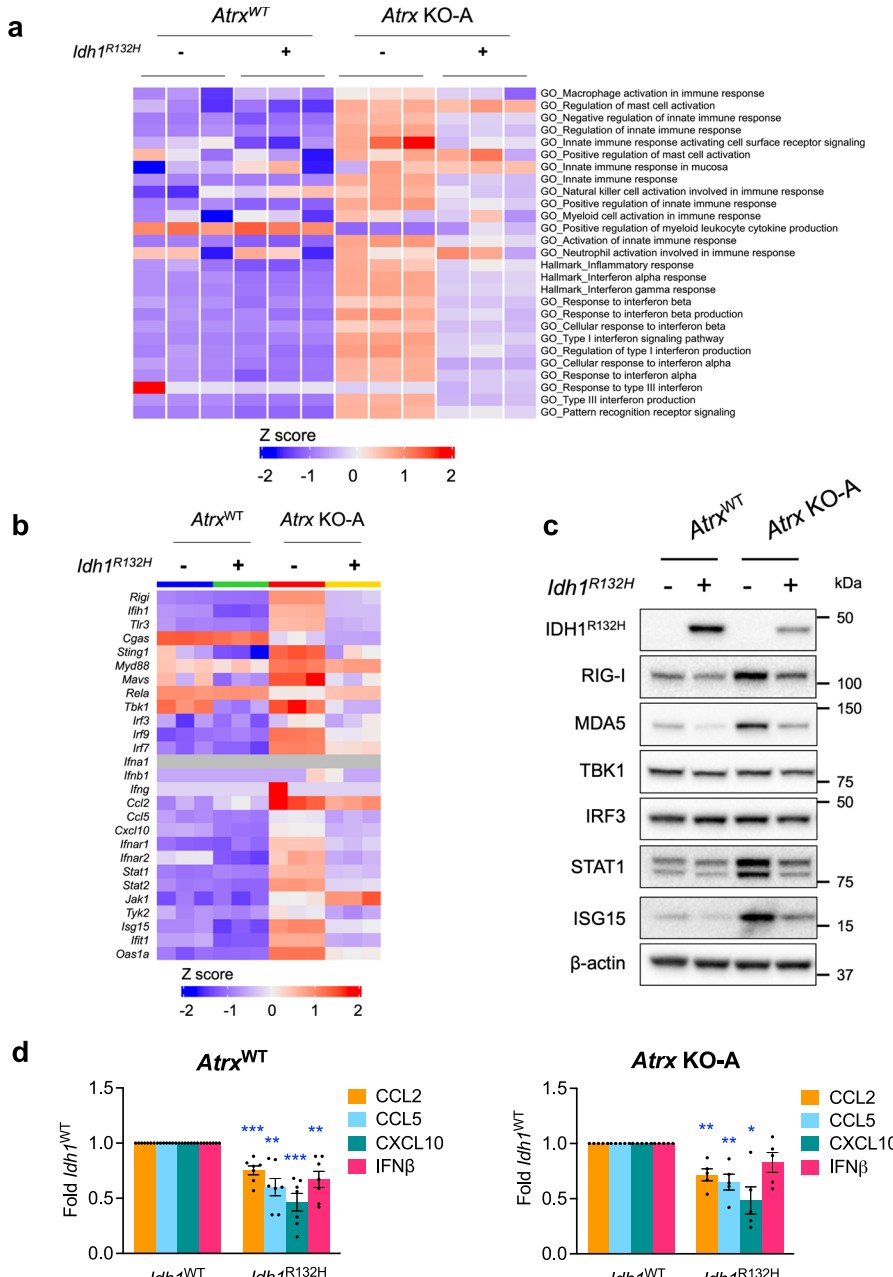

**Fig. 8 | IDH1^R132H co-expression in ATRX-deficient cells dampens baseline innate gene expression. a** Heatmap from ssGSEA showing loss of enrichment of various innate immune-related GO terms in *Atrx* KO-A/ *Idh1^R132H* cells compared to *Atrx* KO-A/ *Idh1^WT* cells. RNA was isolated from cells cultured for 72 h. N = 3 technical replicates per cell line. **b** Heatmap showing differential expression of immune-related genes in CT2A *Atrx*^WT and *Atrx* KO-A cells with or without *Idh1^R132H*. N = 3 technical replicates per cell line. **c** Lysates from CT2A CRISPR control (*Atrx*^WT) or *Atrx* KO-A cells expressing exogenous IDH1^R132H, or empty vector cultured for 72 h, were screened by Western blotting for various innate immune proteins. β-actin serves as the loading control. Representative blots are shown; n = 4 independent experiments. Densitometry values are indicated in Supplementary Fig 14c. **d** Conditioned

media from CT2A CRISPR control (*Atrx*^WT) or *Atrx* KO cells (KO-A) cells expressing exogenous IDH1^R132H, or empty vector (*Idh1^WT*) cultured for 72 h was assayed for antiviral and proinflammatory cytokines using a Legendplex assay kit. Data indicates fold change values normalized to corresponding *Idh1*^WT sample, shown as mean ± SEM. n = 7 independent experiments for *Atrx*^WT (CRISPR control) lines; n = 5 independent experiments for *Atrx* KO-A lines. Asterisks denote significant two-tailed *t*-tests. (*p < 0.05; **p < 0.01; ***p < 0.001). p-values for *Atrx*^WT/ *Idh1^WT* vs *Atrx*^WT/ *Idh1^R132H*: CCL2 – 0.0008, CCL5 – 0.022, CXCL10 – 0.0005, IFNβ– 0.0042; *Atrx* KO-A/ *Idh1^WT* vs *Atrx* KO-A/ *Idh1^R132H*: CCL2 – 0.0066, CCL5 – 0.0083, CXCL10 – 0.0138. Source data are provided as a Source Data file.

## Mutant *IDH1* inhibition partially reverses the immunosuppressive effects of *IDH1^R132H*

Several mutant *IDH* inhibitors are currently being tested for safety and efficacy in the treatment of *IDH1*-mutant gliomas and acute myeloid leukemias, including BAY1436032, a pan-IDH1 inhibitor[35–37]. To evaluate the impact of IDH1^R132H inhibition on the immunological phenotypes detailed above, we treated our panel of CT2A isogenic cells with

1 µM BAY1436032, leading to reduced D2HG levels in both the *Atrx*^WT and *Atrx*-KO lines (Fig. 9a; Supplementary Fig. 15a). Treating IDH1^R132H-expresing cells with BAY1436032 for 72hrs increased baseline expression of RIG-I, MDA5, STAT1 and ISG15, irrespective of the *Atrx* status (Fig. 9b; Supplementary Fig. 15b–d), reverting levels to those seen in the *Idh1*^WT context and demonstrating that crucial components of mutant *IDH1*-associated immunomodulation are reversible. By

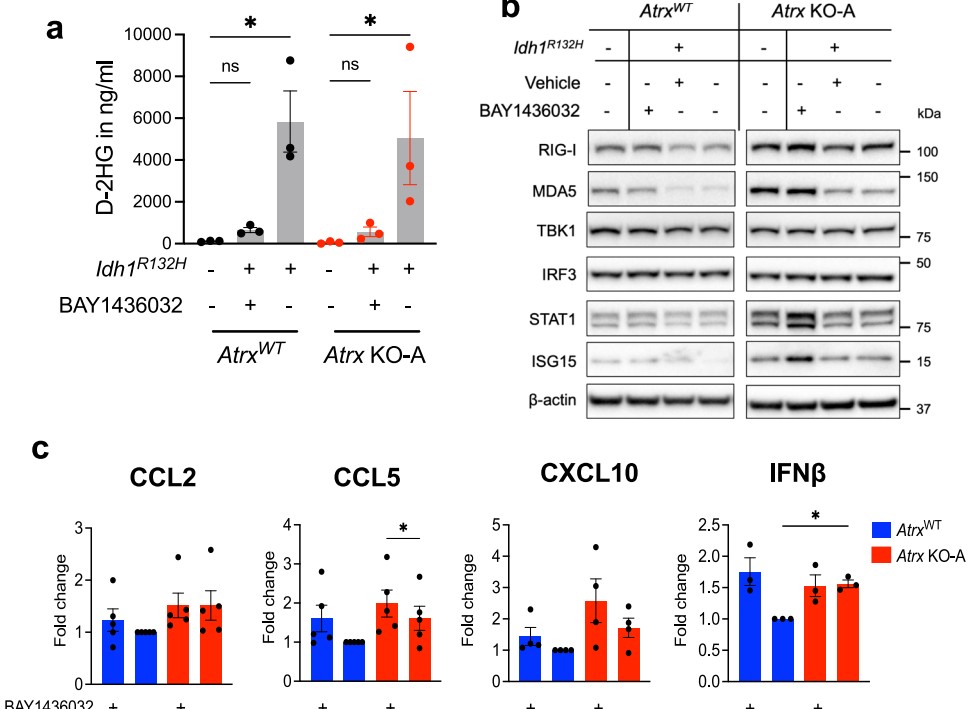

**Fig. 9 | BAY1436032 partially reverses *IDH1*^R132H-mediated immunosuppression.**
**a** D2HG levels in conditioned media from CT2A CRISPR control (*Atrx*^WT) or *Atrx* KO-A cells expressing IDH1^R132H, or empty vector treated with 1 μM BAY1436032 or vehicle every day for 3 days, normalized to total protein. $n = 3$ independent experiments. Data are presented as mean ± SEM. Asterisks indicate significant $p$-values from one-way ANOVA with Sidak post-hoc test. (*$p < 0.05$). **b** Representative Western blots using lysates from CT2A CRISPR control (*Atrx*^WT) or *Atrx* KO-A cells expressing IDH1^R132H or empty vector that were treated with 1 μM BAY1436032 or vehicle every day for 3 days, screened for proteins involved in innate immune signaling. β-actin serves as the loading control. $N = 3$ independent experiments. Densitometry values are indicated in Supplementary Fig 15c. **c** Cytokine/ chemokine levels in conditioned media from CT2A CRISPR control (*Atrx*^WT) and *Atrx* KO-A cells expressing IDH1^R132H treated with 1μM BAY1436032 or vehicle every day for

3 days. Supernatant cytokines were analyzed by cytokine bead arrays for antiviral and proinflammatory cytokines. Fold change values normalized to vehicle treated *Atrx*^WT sample are shown as mean ± SEM. Independent experiments: $n = 5$ for CCL2, CCL5; $n = 4$ for CXCL10; $n = 3$ for IFNβ. One-way ANOVA with Tukey's post-hoc test did not reveal any significant differences between groups for CCL2, and CXCL10. Asterisks indicate significant $p$-values from one-way ANOVA with Sidak post-hoc test (*$p < 0.05$). ns – not significant. **a** $p$-values: *Atrx*^WT/ *Idh1*^WT + Untreated vs *Atrx*^WT/ *Idh1*^R132H + BAY1436032 – 0.995; *Atrx*^WT/ *Idh1*^WT + Untreated vs *Atrx*^WT/ *Idh1*^R132H + Vehicle – 0.0122; *Atrx* KO-A/ *Idh1*^WT + Untreated vs *Atrx* KO-A/ *Idh1*^R132H + BAY1436032 – 0.996; *Atrx* KO-A/ *Idh1*^WT + Untreated vs *Atrx* KO-A/ *Idh1*^R132H + Vehicle – 0.0289. **c** $p$ values: CCL5 for *Atrx* KO-A/ *Idh1*^R132H + Vehicle vs *Atrx* KO-A/ *Idh1*^R132H + BAY1436032: $p = 0.0157$; IFNβ for *Atrx*^WT/ *Idh1*^R132H + Vehicle vs *Atrx* KO-A/ *Idh1*^R132H + Vehicle; $p = 0.0299$. Source data are provided as a Source Data file.

contrast, BAY1436032 treatment did not affect expression of these innate immune proteins in *Idh1*^WT cells (Supplementary Fig. 16a). BAY1436032 also induced pro-inflammatory cytokines, including CCL2, CCL5, CXCL10 and IFNβ in *Atrx*^WT/ *Idh1*^R132H cells to levels equivalent to those of untreated *Atrx*-KO/ *Idh1*^R132H cells (Fig. 9c; Supplementary Fig. 16b). Nevertheless, BAY1436032 treatment of *Atrx*-KO/ *Idh1*^R132H cells failed to significantly increase overall cytokine secretion. Specifically, while CCL5 levels were robustly upregulated, other cytokines, including CCL2 and CXCL10, showed only slight increases that did not reach statistical significance. These findings indicate that the immunomodulatory phenotype induced by mutant *IDH1* in the context of ATRX deficiency is only partially reversible with inhibitory therapy.

### *Atrx*-KO/ *Idh1*^R132H cells retain sensitivity to dsRNA-based immune agonists

To determine the extent to which ATRX-deficient glioma models remain sensitive to dsRNA immune agonism in the context of *IDH1*^R132H, we subjected our isogenic CT2A lines to HMW poly(I:C), monitoring levels of key pathway constituents. Poly(I:C) treatment effectively induced IRF3 and STAT1 phosphorylation and ISG15 expression in IDH1^R132H-expressing cells with concurrent ATRX deficiency (Fig. 10a; Supplementary Fig. 17a). While pIRF3 induction was equivalent in both *Atrx* KO-A and *Atrx* KO-B lines expressing IDH1^WT or IDH1^R132H, pSTAT1 and ISG15 induction was somewhat weakened in *Atrx* KO-A/ *Idh1*^R132H cells. Nevertheless, pSTAT1 and ISG15 were induced at equivalent

levels in both *Atrx* KO-B/ *Idh1*^WT and *Atrx* KO-B/ *Idh1*^R132H lines. Poly(I:C) treatment also increased secretion of cytokines like CXCL1, CCL2, CCL5, CXCL10 and IL6 in *Atrx* KO lines, irrespective of the *IDH1* status (Fig. 10b; Supplementary Fig. 17b). Moreover, the presence of IDH1^R132H did not impair cytokine secretion in either *Atrx* KO-A or *Atrx* KO-B lines upon poly(I:C) treatment, consistent with the trends observed for pIRF3, pSTAT1, and ISG15 described above. These results indicate that IDH1^R132H co-expression in ATRX-depleted cells does not substantively impair dsRNA-mediated induction of the innate immune response. Accordingly, these data further support the notion that the agonizable state of innate immune signaling in ATRX-deficient glioma exists largely independent of *IDH1* mutational status.

## Discussion
Leveraging the immune system in the context of gliomas has been met with relatively diminished return in comparison to its considerable promise in other cancers. This is partially due to the immunosuppressive microenvironment, which has led to tremendous efforts aimed at identifying novel routes for augmenting immune activity and response[38].

While the association between mutations in *ATRX* and *IDH1* has been known for over a decade, the details of their interaction and basis for recurrent co-occurrence in astrocytoma remains unclear. Furthermore, the implication of these mutations on immune-therapeutic responsiveness is understudied. Here, we describe the complex and

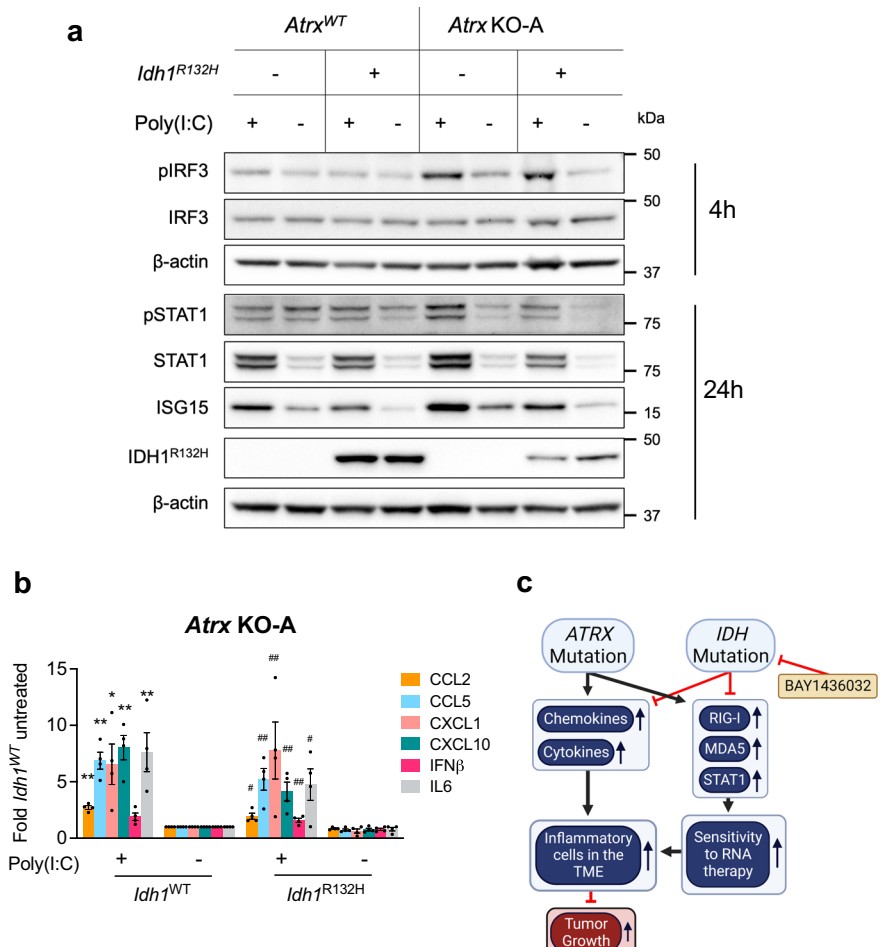

**Fig. 10 | *Atrx* KO/ *Idh1*^R132H cells retain sensitivity to poly(I:C). a** Representative Western blots using lysates from CT2A CRISPR control (*Atrx*^WT) or *Atrx* KO-A cells expressing IDH1^R132H or empty vector that were treated with 10 µg/ml poly(I:C) HMW for 4 h or 24 h and screened for proteins involved in innate immune signaling. *N* = 3 independent experiments. **b** Cytokine levels in conditioned media from CT2A *Atrx* KO-A cells expressing IDH1^R132H or empty vector treated with poly(I:C) HMW for 24hrs. Supernatant cytokines were analyzed by cytokine bead arrays for antiviral and proinflammatory cytokines. Fold change values normalized to untreated *Atrx* KO-A/ *Idh1*^WT sample are shown as mean $\pm$ SEM. *n* = 4 independent experiments. Asterisks and hashtags indicate significant *p*-values from one-way ANOVA with

Tukey's post hoc test comparing poly(I:C) HMW with untreated samples for *Atrx*-KO-A/ *Idh1*^WT line (*) or *Atrx*-KO-A/ *Idh1*^R132H (#) (*, #*p* < 0.05; **, ##*p* < 0.01; ***, ###*p* < 0.0001). *p*-values for *Atrx* KO-A/ *Idh1*^WT + Untreated vs *Atrx* KO-A/ *Idh1*^WT + poly(I:C) HMW (shown as *): CCL2 − 0.0017, CCL5 − 0.0014, CXCL1 − 0.0347, CXCL10 − 0.0028, IL6 − 0.0099; *p*-values for *Atrx* KO-A/ *Idh1*^R132H + Untreated vs *Atrx* KO-A/ *Idh1*^R132H + poly(I:C) HMW (shown as #): CCL2 − 0.0357, CCL5 − 0.0029, CXCL1 − 0.0015, CXCL10 − 0.0039, IFNβ − 0.0012, IL6 − 0.0179. **c** Cartoon summarizing the effect of *ATRX* mutations on baseline and dsRNA-mediated innate signaling and its interplay with *IDH* mutations in gliomas. Source data are provided as a Source Data file.

contrasting impact of these disease-defining molecular alterations on the immune microenvironment and characterize strategies for therapeutic targeting (Fig. 10c).

Mutations in *IDH* contribute to the overall immunosuppressive microenvironment characteristic of gliomas[25,32,39]. However, the precise role of molecular alterations co-occurring with *IDH* mutations, including ATRX inactivation, are only beginning to be elucidated[23–26,32,39]. We demonstrated that *ATRX*-mutant astrocytomas display a more pro-inflammatory gene expression profile compared to oligodendroglioma counterparts. Suggesting a connection between ATRX loss and immune-modulatory behavior, pro-inflammatory gene expression profiles are enriched among *ATRX*-mutant low-grade gliomas. This was apparent through analysis of both TCGA-LGG bulk RNAseq data and RNAseq analysis of our genetically engineered murine and human glioma models. In the process of developing our orthotopic models (both CT2A and RCAS/Ntv-a), we observed prolonged survival among animals bearing ATRX-inactivated tumors, which was associated with an increase in T-cell infiltration in vivo, and enhanced innate immune signaling and production of cytokines and chemokines indicative of an activated immune response in vitro.

Interestingly, this survival phenotype was observed to a lesser extent when ATRX-deficient cells were implanted into nude animals, implicating immune microenvironmental engagement as a fundamental driver of this phenotype. Supporting a role for the tumor microenvironment, T-cell depletion studies in immunocompetent mice bearing *Atrx*^WT or *Atrx* KO tumors have implicated a role for CD8 + T cells in prolonging survival. This genetic and transcriptional data derived from ATRX-deficient human and murine systems, coupled with in vitro upregulation of RIG-I and MDA5 and increased cytokine production of CCL2, CCL5, CXCL10, and IFNβ depicted a robust pro-inflammatory phenotype and stimulation of the innate arm of the immune system. In total, these findings link ATRX loss with a pro-inflammatory state.

ATRX loss has been shown to contribute to immune-modulatory behavior through its association with the ALT phenotype and subsequent production of extrachromosomal telomere repeats (ECTR). When ATRX is intact, any extracellular DNA activates the cGAS-STING pathway[40,41], triggering a type 1 interferon response with production of IFNβ, inhibiting cell growth, and/or promoting cellular elimination. However, recent work has shown that *ATRX*-mutant and ALT-positive

cells may confer defective cytosolic DNA sensing by inactivating the cGAS-STING pathway, precluding the type 1 interferon response[40,42]. Similar to the downstream effects of the cGAS-STING pathway, activation of dsRNA sensors such as RIG-I and MDA5 ultimately leads to production of type I interferons and other pro-inflammatory cytokines[43]. We observed an induction in RIG-I and MDA5 in both our human and mouse ATRX-deficient glioma models, indicative of enhanced sensitivity to dsRNA. The precise molecular mechanisms underlying this "primed" cellular state remain unclear. ATRX is a well-established epigenetic regulator and as such, its loss could directly impact the expression of relevant immune pathway genes in the appropriate developmental context, poising affected cells for integrated transcriptional responses to dsRNA. ATRX inactivation has also been repeatedly linked to derepression of endogenous retroviral (ERV) elements[44,45], which could lead to increased cytoplasmic dsRNA, engagement of antiviral sensing pathways, and interferon responses[46–48].

Regardless, we reasoned that this sensitized cell state might be effectively leveraged therapeutically. To this end, we investigated the response of ATRX-deficient cells to the dsRNA agonist poly(I:C), which led to increased secretion of pro-inflammatory cytokines indicative of immune stimulation. These findings indicate an important mechanism of dsRNA-mediated innate immune sensitivity in ATRX-deficient gliomas, distinct from the effects of ATRX loss on dsDNA-mediated immune suppression noted in previous studies. While this phenotype was observed across murine and human models, the extent of in vivo response remains to be seen and will be the foundation of future studies. Specifically, understanding how dsRNA agonists can mount an enhanced immune response and whether this confers a therapeutic benefit under in vivo conditions will be paramount. Indeed, various treatment approaches, including oncolytic virotherapies grounded in the induction of RNA sensing and stimulation of the innate immune system have shown promise for several cancers including GBM[11,12,14]. For instance, recent work utilizing recombinant nonpathogenic polio-rhinovirus chimera (PVSRIPO) has shown improved survival for treated patients with recurrent glioblastoma[11].

Considering their frequent co-occurrence as well as their respective roles in immune response, understanding how *ATRX* and *IDH* mutations influence immune state in combination is warranted. We demonstrated that among ATRX-deficient cell models, the presence of the *IDH1* mutation has negligible impact on proliferation; however, upon intracranial implantation, mutant *IDH1* significantly increases tumor aggressiveness, as indicated through a shorter survival and greater tumor penetrance. Interestingly, while mutant *IDH1* did not impact the enrichment of various innate immune-related gene sets in the context of wildtype *ATRX*, it did depress the induction of the innate immune response when ATRX was lost. Similarly, many of these genes were downregulated in response to mutant *IDH1*, reverting their expression to the baseline levels seen in *ATRX*^WT cells, conferring an overall immunosuppressive impact, and dampening of the pro-inflammatory effect of ATRX loss. Nevertheless, despite this mutant *IDH1*-mediated immunosuppressive phenotype, ATRX-deficient cells remained capable of mounting a robust innate response upon poly(I:C) treatment, a key finding of therapeutic relevance.

It should be noted that most studies examining the combined pathogenic impact of *IDH* mutation and ATRX inactivation thus far have addressed these molecular alterations concurrently[23,24], whereas in the context of human disease, IDH mutation is thought to arise first followed by the sequential acquisition of ATRX deficiency[49,50]. With both IDH1 and ATRX abnormalities, each exerting independent epigenetic effects on cellular physiology, it is possible that their sequential emergence and interval "priming" of chromatin and transcriptional landscapes could alter ultimate phenotypic expression, accounting for discrepancies in the literature. In this way, the precise timing and order of mutational acquisition could significantly impact therapeutically

tractable sensitivities conferred by ATRX deficiency in glioma cells. For instance, one study investigating the effects of ATRX loss in the context of *IDH* and *TP53* mutations concluded that ATRX loss confers an immunosuppressive state as evidenced by upregulated secretion of relevant cytokines and chemokines (CXCL8, IL6, CXCL3, and CSF2) and increased tumor growth[24]. It is worth noting that the in vivo arm of that study relied on GL261, a model known for high baseline immunogenicity[51], whereas both our CRISPR and RCAS/Ntv-a models better reflect the low baseline immunity seen in the human context. Another study that developed murine glioma models based on *ATRX* KO followed by mutant IDH1 overexpression described upregulated pro-inflammatory gene programs and Nfkb1 signaling as characterized by single cell RNAseq and ATAC-seq[23]. However, the observed accumulation of immunosuppressive M2 macrophages in this model complicated final conclusions regarding the impact of ATRX loss.

Several strengths and weaknesses are noted in our models. The CT2A glioma line extensively leveraged in this work may already harbor intrinsic tumor-related molecular mechanisms independent of ATRX loss and mutant *IDH1* that may impact immune microenvironmental engagement. That being said, our RCAS/Ntv-a *Atrx* KO model, though lacking mutant *IDH1*, closely mirrors the natural progression of low-grade gliomas in humans and corroborates the immunological effects of ATRX loss noted in human gliomas and the CT2A *Atrx* KO model. While the aforementioned studies examining the immunologic impact of ATRX deficiency relied heavily on RNA-level data, our work integrated protein and gene set level data, along with functional changes in vivo. Importantly, this study adds to prior work demonstrating the hypersensitivity of ATRX-deficient cells to dsRNA-based innate immune agonism. These findings are further supported by our analysis of human TCGA data, showing that combined ATRX loss and *IDH* mutation in astrocytomas yield a pro-inflammatory phenotype relative to *IDH* mutation alone.

In light of the mitigating effects conferred by *IDH* mutation on the pro-inflammatory phenotype of ATRX deficiency, we considered the possibility that mutant *IDH1* inhibition could restore immune pathway engagement in our experimental model. Indeed, when the pan-mutant *IDH1* inhibitor, BAY1436032[35–37] was administered to *Atrx-KO/Idh1*^R132H lines, we found increased expression of RIG-I, MDA5, STAT1, and ISG15, relative to vehicle-treated controls. These trends suggest that mutant *IDH1*-mediated suppression of innate immune response in ATRX deficient lines can be at least partially rescued by BAY1436032. However, the full impact of mutant IDH1 inhibition and dsRNA agonism on immune cell recruitment remains to be determined and will be the foundation of future studies. Based on the in vitro data presented, investigations into therapeutics seeking to exploit the ATRX-deficient pro-inflammatory phenotype appear promising.

## Methods
### Cell lines
Human M059J cells were obtained from ATCC (Manassas, VA). Mouse CT2A cells were a kind gift of Peter Fecci (Duke University Medical Center). Human 293FT cells were obtained from Thermo Scientific (Waltham, MA). M059J cells were maintained in DMEM/F12 medium containing 2.5 mM L-glutamine, 15 mM HEPES, 0.5 mM sodium pyruvate, 1.2 g/L sodium bicarbonate and 10% fetal bovine serum (FBS). CT2A and 293FT cells were maintained in DMEM (high glucose) containing 10% FBS. All cell lines were cultured without antibiotics. Cell lines were routinely tested for mycoplasma contamination using the MycoAlert® kit (Promega, Madison, WI).

### Generation of CRISPR-Cas9-mediated *ATRX* KO cell lines
For generating *ATRX* KO cell lines, single guide RNAs (sgRNAs) targeting exon 9 of human *ATRX* or mouse *Atrx* (Supplementary Table 1) were cloned into CRISPR-Cas9 lentiCRISPR v2 vector (gift from Feng Zhang, Addgene plasmid #52961). Lentivirus was produced in

293FT cells. CT2A cell line was transduced with a pair of lentiviruses targeting two regions of exon 9 simultaneously to ensure efficient gene knockout, while M059J cell line was transduced with lentivirus targeting a single region of exon 9. Two days after transduction, cells were selected with puromycin for 5–7 days, and gene knockout confirmed by sequencing and western blot. Alternatively, lines were subjected to single cell cloning. A polyclonal population of ATRX-KO M059J cells and single cell clones isolated from CT2A Atrx-KO cells were used for experiments. ATRX knockout in these final reagents was confirmed by sequencing and immunoblotting. Since ATRX loss is associated with DNA damage and genomic instability[52], only early passage ATRX-KO cell lines were used for experiments.

### Generation of $Idh1^{R132H}$ cell lines
IDH1[R132H]-expressing CT2A lines were generated using a retroviral MSCV-$Idh1^{R132H}$ construct, or its corresponding empty vector control. Lentiviruses produced in 293FT cells were transduced into CT2A CRISPR control and Atrx KO clones, KO-A and KO-B lines. All cells were transduced twice to improve efficiency. Cells were allowed to expand for a week after transduction and expression of IDH1[R132H] was confirmed by immunoblotting and immunofluorescence.

### In vivo CT2A model
All animal study protocols were approved by and performed in accordance with Duke Institutional Animal Care and Use Committee (IACUC) guidelines. Female nude (JAX# 002019) or C57BL/6 mice (JAX #000664) aged 8 to 10 weeks were purchased from Jackson Laboratories (Bar Harbor, ME). Animals were housed in filter-top cages in Thoren units within the Duke University Cancer Cell Isolation Facility (CCIF) with 12 h light-dark cycles at temperature of 21 °C (±3 °C) and a relative humidity of 30–70%. Food and water were provided *ad libitum*. CT2A cells were intracranially implanted into the right caudate nucleus (Implant coordinates: −2.0, 0.5, 3.7/ 3.4) using an inoculation volume of 5µl. Mice were implanted with an inoculum of 150,000 cells suspended in 3% methylcellulose (30,000 cells/µl). Animals were monitored daily and euthanized upon development of neurological symptoms or signs of distress. Brains from all animals were harvested at necropsy. Brains were fixed overnight in 10% formalin, paraffin-embedded and sectioned (5µM thickness) for hematoxylin-eosin (H&E) and immunostaining.

For CD4+ and CD8 + T-cell depletion experiments, C57BL/6 mice (*n* = 10 mice per group) received 250 µg per dose of rat IgG2B isotype control (BioXcell, Cat. no. BE0090), anti-mouse CD4 antibody (BioXcell, Cat. no. BE0003) or anti-mouse CD8 antibody (BioXcell, Cat no. BE0061) intraperitoneally 4 days prior to implant. Mice were then implanted with CT2A CRISPR control (Atrx[WT]) or Atrx KO-B cells on day 0 and were administered with the respective antibodies on the day of implant and every 3–4 days thereafter as per schema in Supplementary Fig. 9. Animals were euthanized upon development of neurological symptoms or signs of distress and brains from all animals were collected at necropsy. Antibodies used in this study are listed in Supplementary Table 2.

### RCAS model – in vivo and in vitro
All animal study protocols were approved by and performed in accordance with protocols approved by the MD Anderson Cancer Center (MDACC) Institutional Animal Care and Use Committees. J12 (nestin-Tva bearing, JAX #003529) mice were generously provided by Dr. Eric Holland and ATRX $^{fl/fl}$, (Atrx[tm1Rjg], MGI #3528480) mice were acquired from Dr. David Picketts. Initial generation of RCAS tumors was done via injection of RCAS bearing cells into 1-5 day old mice. Xenograft experiments were conducted in mice at 8 to 10 weeks of age. Animals were housed in filter-top cages in Tecniplast units within the MD Anderson North Campus Animal Facility, with a 12 h light cycle from 6:00am–6:00 pm, at temperature of 22.2 °C (± 2 °C) and a relative humidity of 40–55% (Set point 45%). Food and water were provided *ad libitum*. These mice were crossed, and pups genotyped by PCR from tail-derived genomic DNA. The RCAS-PDGFα-shp53 and RCAS-Cre constructs were generously provided by Dr. Eric Holland. DF1 cells were transfected with the different RCAS retroviral plasmids using FuGENE 6 Transfection reagent (Promega, Cat. no. E2691), accordingly to the manufacturer's protocol. For RCAS-mediated gliomagenesis, newborn mice of both sexes were injected intracranially with $4 \times 10^5$ DF1 cells per mouse, combining RCAS-PDGFa-shp53 and RCAS-Cre expressing cells at a 1:1 dilution. For xenografting experiments, adult male mice were used, as the cell line was male in origin. Mice were anaesthetized by 4% isoflurane and then injected with a stereotactic apparatus (Stoelting) as previously described[28]. After intracranial injection, mice were checked until they developed symptoms of disease (lethargy, poor grooming, weight loss, macrocephaly). For generation of cell lines, tumors from RCAS-injected mice were harvested, and mechanically and chemically homogenized. Single cell suspensions were cultured in NeuroCult Basal Medium containing NeuroCult Proliferation Supplement, 20 ng/ml EGF, 10 ng/ml basic FGF, 2 µg/ml heparin (Stemcell Technologies), 50 units/ml penicillin, and 50 µg/ml streptomycin (Thermo Fisher Scientific).

### Immunohistochemistry
IHC was performed on 5 µM formalin-fixed, paraffin embedded (FFPE) brain sections. IDH1[R132H] staining was performed by Duke Pathology Research Histology and Immunohistochemistry Laboratory, while ATRX and CD3 staining for the RCAS/Ntv-a model was performed in house in our laboratory at UT MD Anderson. ATRX IHC for CT2A model and CD45 and F4/80 IHC for both the CT2A and RCAS/Ntv-a models were performed at Histowiz, Inc (Brooklyn, NY). Antibodies used for immunostaining are listed in Supplementary table 2.

Immunostaining was performed using the Leica Bond RX automated stainer (Leica Microsystems) at Histowiz, Inc using a fully automated workflow, which included deparaffinization, epitope retrieval and incubation with peroxide block buffer. This was followed by incubation with respective primary and secondary antibodies according to the manufacturer's protocol. Slides mounted with coverslips were visualized and imaged using a Leica Aperio AT2 slide scanner (Leica Microsystems) at 40X. Immunostaining for IDH1[R132H], ATRX and CD3 performed at Duke and MD Anderson followed a similar manual workflow. Slides were visualized and imaged using an Echo Revolve Microscope, using the Echo Pro software (Duke University) or Keyence BZ-X810 microscope, using the BZ-X800 viewer acquisition software (UT MD Anderson). QuPath software was used for IHC quantitation.

### Growth assay
Cell growth was monitored using Cell Titer 96 AQueous One proliferation assay (MTS) (Promega, Madison, WI) as per manufacturer's instructions. Cells were plated onto 96-well plates as triplicates at the following densities: CT2A – 2000 cells/ well; M059J – 5000 cells/ well. Growth was measured at 24, 48, 72 and 96 h after cell plating by combining cells in media with CellTiter 96 Aqueous Solution reagent, incubating at 37 °C for 3 h and measuring absorbance at 490 nm. Wells containing media only were included for background subtraction.

### Cell treatments
Innate immune agonists obtained from Invivogen (San Diego, CA) include poly(I:C) LMW (Cat no: tlrl-picw) and poly(I:C) HMW (Cat no: tlrl-pic). BAY1436032 was obtained from MedChem Express (Monmouth Junction, NJ) and maintained as 10 mM stocks in 100% ethanol. Cell lines treated with 10µg/ml poly(I:C) LMW or HMW for 4 h or 24 h were harvested for protein extraction at either 4 h or 24 h timepoints, while conditioned media was collected at the 24 h timepoint. Cells treated with 1 µM BAY1436032 or with ethanol (vehicle control) every

day for 3 days were harvested 24 h after the final dose for protein extraction, while conditioned media collected at the same timepoint was used for measuring D2HG and cytokine levels.

## Protein extraction and Western blotting
Monolayer cell cultures were washed in ice-cold 1X PBS twice and lysed in RIPA buffer containing Halt Protease and Phosphatase Inhibitor cocktail (Thermo Scientific) and Benzonase (Milliprore Sigma, Burlington, MA). Cell lysates were incubated on ice for 15 mins, followed by centrifugation at ~14,000 × g for 15 mins at 4 °C. Supernatants were used to determine protein concentrations by Bradford assay. Twenty micrograms of protein were used in gel electrophoresis and western blotting. PVDF membranes were incubated with primary antibody overnight, followed by secondary antibody for 1 hour and processed for protein detection using SupersignalTM West Pico PLUS (Thermo Scientific) or Immobilon Western chemiluminescent substrate (Millipore Sigma). Images were captured using a Biorad Chemidoc MP imaging system and analyzed using ImageLab 6.1 software (Biorad, Hercules, CA). Primary and secondary antibodies used in this study and the respective dilutions are listed in Supplementary Table 2. Uncropped blot images are available in the source data file.

## Cytokine/chemokine analysis
Cytokine levels in conditioned media from cells were measured using Legendplex (Biolegend, San Diego, CA) as per the manufacturer's instructions on a BD Fortessa X-20 flow cytometer. For CT2A and RCAS/Ntv-a samples, the mouse anti-virus response panel (Cat #: 740621) was used. For M059J samples, the human anti-virus response (Cat #: 740390) and pro-inflammatory chemokine panels (Cat #: 740984) were used. Samples were subjected to a 1:10 dilution to quantify levels of CCL5 and CXCL10 within the linear detection range. Legendplex Data Analysis software (https://legendplex.qognit.com) was used to calculate cytokine concentrations. Only cytokines with detectable induction upon treatment are shown.

## Flow cytometry analysis
Flow cytometry analysis on tumor bearing hemispheres harvested on day 14 post implant from C57BL/6 mice implanted with CT2A CRISPR control, *Atrx* KO-A or KO-B cells was performed as previously reported[13]. Tumor-bearing hemispheres were minced and dissociated using Liberase and DNaseI to generate single cell suspensions. Any remaining tissue pieces were dissociated through a 70 µm cell strainer, washed in HBSS and subjected to myelin removal after overlaying on 20% Percoll (MP Biomedicals) solution in HBSS. Single cell suspensions stained with Zombie-Aqua (Biolegend, 1:500), were then stained with the following antibody panels (all Biolegend unless otherwise specified at 1:100 dilution): panel 1: CD45.2-BUV395 (BD Biosciences), CD3-FITC, CD19-FITC, NK1.1-BV421, CD11b-BV711, and Ly6G-PE; panel 2: CD45.2-BUV395, CD3-PE, CD4-FITC, CD8-BV421, PD1-PEcy7, and Tim3-BV711 (Antibodies are listed in Supplementary Table 2). RCAS tumors implanted in Ntv-a, *Atrx*^fl/fl mice (C57BL/6 background) were harvested 14 days post-implantation and processed as detailed above. RCAS tumors were stained with the following panel: CD45-BUV395, CD8a-BV421, NK1.1-BV605, IA/IE-BV786, CD4-FITC, Ly6c-PerCP-Cy5.5, CD3-PE, F4/80-APC, Ly6G-PE-Cy7, CD11b-APC-Cy7, and Zombie Aqua for live dead (Antibodies are listed in Supplementary Table 2). Antibody-stained cell suspensions were analyzed using a BD Fortessa X-20 flow cytometer. Data were analyzed using Flow-Jo v10.8.1 (BD Biosciences). Gating strategies are included in Supplementary Fig. 6.

## D-2-hydroxyglutarate (D2HG) measurements
Conditioned media or cell pellets resuspended in PBS were used to quantify D2HG by liquid chromatography/ electrospray ionization tandem-mass spectrometry (LC-ESI-MS/MS) as published previously[53–55], with modifications to accommodate equipment and sample matrix. D2HG concentrations in cell pellets were normalized to total protein.

## Materials
D-2-HG and Diacetyl-L-tartaric anhydride (DATAN) were from Sigma. Racemic mixture of L- and D-2-HG-d4 was used as internal standard and was in-house made by reduction of α-ketoglutarate-d6 (Sigma/Isotec) by NaBH$_4$ (Sigma) in anhydrous MeOH (Sigma). Other reagents and solvents were of analytical grade.

## Sample preparation/derivatization
To 10 µL of conditioned media or cell suspension in PBS, 10 µL of 20 µg/mL of L/D-2-HG-d4 (internal standard) in methanol was added, vigorously agitated in FastPrep (Thermo), and mixture evaporated to dryness under gentle stream of nitrogen at 65 °C. The dry residue was treated with 50 µL of 50 mg/mL of freshly prepared DATAN in dichloromethane/glacial acetic acid (4/1 by volume) and heated at 75 °C for 30 min. After drying (65 °C, 1 h) the residue was reconstituted in 50 µL LC mobile phase A (see below) for LC/MS/MS analysis.

## LC/MS/MS analysis
Instrument: Agilent 1200 series HPLC and Sciex/ Applied Biosystems API 5500 QTrap. Mobile phase: (A) water, 3% acetonitrile, 2 mM ammonium hydroxide, pH adjusted to 3.6 by formic acid; (B) acetonitrile. Analytical column: Agilent Eclipse Plus (C$_{18}$, 1.8 µm, 150 × 4.6 mm), at 50 °C. Elution gradient: 0–3 min 0.5%B, 3–4 min 0.5-90%B, 4–5 min 90-0.5% B. Run time: 11 min. Injection volume: 5 µL. Mass spectrometer parameters (voltages, gas flow, and temperature) were optimized by infusion of 100 ng/mL of analytes in mobile phase at 10 µL/min using Analyst 1.6.2 software tuning module. The Q1/Q3 (m/z) transitions monitored: 363/147 (D-2-HG) and 367/151 (L/D-2-HG-d4).

## Calibration and quantification
A set of calibrator samples in fresh media or PBS was prepared by adding appropriate amounts of pure D-2-HG in 0.0322 − 100 µg/mL range. The calibration samples were analyzed alongside the experimental samples. Accuracy acceptance criteria was 85% for each but the lowest level (0.032 µg/mL, 80%, LLOQ). The obtained calibration curve was linear ($r^2 = 0.999$). Analyst 1.6.2 software was used for data acquisition, integration of the chromatograms, calibration curve calculation, and quantification of the study samples.

## RNA extraction and gene expression analysis
RNA was extracted from monolayer cell cultures using miRNeasy Mini kit (Qiagen, Germantown, MD) according to manufacturer's instructions. mRNA sequencing on CT2A *Atrx* KO lines and M059J *ATRX* KO lines was performed by Genewiz Azenta Life Sciences (South Plainfield, NJ) as per their workflow. cDNA libraries were sequenced using a HiSeq 4000 (Illumina) to generate 150 bp pair-end reads. mRNA sequencing on CT2A MSCV EV control or *Idh1*^R132H lines was performed by the Sequencing and Genomics Technologies Core Facility at Duke University. cDNA libraries generated using the Kapa stranded mRNA Kit (Roche Kapa Biosystems, Indianapolis, IN) were pooled to equimolar concentrations and sequenced on the NovaSeq 6000 S-Prime flow cell to produce 100 bp paired-end reads. Sample fastq file's quality metrics were assessed via FastQC v0.11.9 and MultiQC v1.11. Next, the paired-end reads were aligned to the GRCh38 human reference genome or GRCm38.p6 mouse reference genome via STAR v2.7.2b with default parameter settings. Subsequently, post-alignment quality metrics were assessed via the log file output of STAR. Quantification and generation of the raw counts matrix was performed via featureCounts v1.6.3. Differential expression was calculated by normalizing the raw counts matrix using DeSeq2 R package v1.38.2.

For RCAS tumor and cell line sequencing, RNA extraction was performed with Qiagen RNeasy Plus kit, according to manufacturer's instructions. Subsequent processing and sequencing occurred at the

MD Anderson Advanced Technology and Genomics Core (ATGC). Following Agilent BioAnalyzer quality assessment, libraries were generated using Truseq library preparation kits (Illumina) and samples run on the HiSeq4000 platform in a 76 bp pair-end sequencing format. The raw fastq files were subjected to FASTQC analysis for quality control analysis, followed by alignment to mouse mm9 genome using RNASTAR (version 2.7.8a). Raw transcript counts were generated using HTseq-count tool (version 0.9.1), followed by Principal Component Analysis (PCA). Differentially expressed genes (DEGs) were calculated using limma-voom (version 3.44.3) and were later subjected to Gene Set Enrichment Analysis (GSEA) analysis using a desktop version of the analysis tool.

To perform single sample gene set enrichment analysis (ssGSEA), log normalized TPM counts were input into GSVA v1.47.0 with method set to "ssgsea" and a list of custom gene sets. Enrichment scores were then Z-score transformed via the scale function in R, with the matrix transposed to scale across sample IDs. Heatmaps were generated via the package ComplexHeatmap v2.14.0.

## TCGA data
Gene mutation data from the CNS/Brain TCGA LGG PanCancer Atlas Study from cBioportal was used to generate an Oncoprint output for a select number of immune genes. RSEM batch normalized counts (RNAseq) for 410 samples from the same dataset were stratified by *IDH* and *ATRX* mutation status, and 1p/19q codeletion status. Log normalized counts was subject to ssGSEA as described above for cell line RNAseq data.

## scRNA-Seq data extraction & analysis
scRNA-Seq public datasets *IDH*-mutant astrocytoma (GSE89567)[26] and oligodendroglioma (GSE70630)[27] were downloaded from NCBI GEO. Seurat's v4.3.0 standard pre-processing workflow was used with clustering resolution determination aided by the R package clustree v0.5.0. Immune and tumor canonical marker genes were utilized along with differential expression lists to annotate each dataset's clusters. Clusters annotated as immune cells or tumor cells were identified and integrated using Seurat's integration workflow. Differential gene expression analysis was then performed via the FindAllMarkers function. Differential expression results were ranked by log2FC and then input into the GSEA R package fgsea v1.22.0 along with the Hallmark gene set collection.

## Statistics and reproducibility
Quantitative results for all experiments are expressed as mean ± SEM. The number of samples and biologically independent experiments or technical replicates and associated statistical tests are indicated in the corresponding figure legends. For blots and IHC images showing representative images, each experiment or staining was repeated atleast three times independently showing similar results. Data plotting and statistical analysis were performed using GraphPad Prism v9. *p*-values less than 0.05 are considered statistically significant and significant differences are indicated by asterisks (*) or hashtags (#). Individual data points are shown for all graphs.

## Reporting summary
Further information on research design is available in the Nature Portfolio Reporting Summary linked to this article.

## Data availability
All data generated during this study are included in this published article, supplementary data files or in the source data file. All transcriptional data generated in this study have been deposited in Gene Expression Omnibus (GEO) under accession numbers: GSE228242 (CT2A *Atrx*-KO cell lines), GSE228181 (CT2A *Atrx*-KO/ *Idh1* mutant cell lines), GSE228243 (M059J *ATRX*-KO cell lines), GSE231830 (RCAS *Atrx*- KO cell lines) and GSE231831 (RCAS *Atrx*-KO tumors). The following publicly available datasets were also analyzed for this study - CNS/ Brain TCGA LGG PanCancer Atlas Study from cBioportal and scRNA-Seq datasets from GEO - *IDH*-mutant astrocytoma (GSE89567) and oligodendroglioma (GSE70630). Correspondence and requests for materials should be addressed to David M. Ashley and Jason T. Huse. Source data are provided with this paper.

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

## Acknowledgements

We thank Feng Zhang for providing the CRISPR-Cas9 lentiCRISPR v2 vector (Broad Institute, Boston, MA). We thank Duke's Sequencing and Genomic Technologies Shared Resource for providing the mRNA-sequencing service, Duke's Brain Tumor Center Biorepository and Database, Substrate services core research support and Duke's Pathol-ogy Research Histology and Immunohistochemistry Laboratory for IHC staining. We thank the MD Anderson Genomics Core Facility for mRNA-sequencing on the RCAS model. This work was supported by NIH R01 CA255788 (D.M.A, J.T.H and M.C), R01 CA240338 (J.T.H.) and American Cancer Society's Research Scholar's Grant, RSG-16-179-01-DMC (J.T.H.). Research reported in this publication was supported by the National Center for Advancing Translational Sciences of the National Institutes of Health under Award Numbers TL1TR003169 and UL1TR003167 (B.T.W) and NCI Cancer Center Support Grant (CCSG), P30CA014236 (I.S.). The content is solely the responsibility of the authors and does not neces-sarily represent the official views of the National Institutes of Health. We would like to acknowledge V Foundation for Cancer Research, Jewish Communal Fund Grant and Brain Tumor Research Charity Grant for funding provided to D.M.A. and the Ben and Catherine Ivy Foundation and the Brockman Foundation for funding provided to J.T.H.

## Author contributions

Conceptualization and design: S.H., B.T.W., M.S.W., M.C.B., M.L.B., D.M.I., S.D., S.T.K., M.G.C., J.T.H., D.M.A. Development of methodology: S.H., B.T.W., M.S.W., M.C.B., M.L.B., D.M.I., S.D., S.T.K., Y.H., J.T.H., D.M.A. Acquisition of data: S.H., B.T.W., M.C.B., M.L.B., D.M.I., K.R., R.F., J.H., S.D., E.A.G., A.B., A.M., C.F., M.S., M.R., I.S., S.T.K., Y.H. Analysis and interpretation of data: S.H., B.T.W., C.J.P., M.S.W., M.C.B., M.L.B., D.M.I.,

S.D., K.S., G.Z., P.B.M., H.R., S.T.K., Y.H., M.G.C., J.T.H., D.M.A. Writing, review and/or revision of manuscript: S.H., B.T.W., C.J.P., M.S.W., M.C.B., M.L.B., M.G.C., J.T.H., D.M.A. Administrative, technical, or material support: S.H., B.T.W., M.L.B., J.T.H., D.M.A. Study supervision: J.T.H., D.M.A.

## Competing interests

The authors declare no competing interests.
