## [Peer Review File · Nature Communications]

Reviewers' Comments:

Reviewer #1:

Remarks to the Author:

The manuscript "Interplay between ATRX and IDH1..." submitted by Hariharan et al describes studies addressing the role that ATRX loss and IDH1 mutation alone and in combination play in regulating the innate immune response. The key findings of the study are that loss of ATRX expression leads to increased innate immune signaling and cytokine secretion in response to dsRNA stimuli, which in turn appears to be masked by co-expression of mutant IDH1 as is noted in lower-grade astrocytoma. The key take home message of this study is that pharmacologic suppression of IDH1 may in turn reveal the effects of ATRX loss and in turn increase therapeutic response to dsRNA-based therapies. The latter finding is novel, significant, and noteworthy.

There are a number of issues that if addressed would improve the credibility of this work. First, technically, it would be very useful to include protein size markers and quantitation in all Western blots displayed. Size markers would help validate the antibodies used while quantitation is required to support claims of differences in expression, especially in cases such as in Fig 7C and 8b where the suggested changes are not immediately apparent.

Second there are some technical issues with the cells used in this study. A key finding here is that loss of ATRX expression significantly slows the intracranial growth of cells but only in immunocompetent animals. This is clearly shown in both CT2A and RCAS models. A point glossed over here, however, is that loss of ATRX in CT2A cells also greatly slows the growth of these cells in vitro (Fig 1e), although this apparent >50% decrease is described as "modest" on page 6 and "negligible" in the discussion on page 14. What was the exact doubling time of these cultures and is it significantly different in these groups? If it is significantly longer in the ATRX KO CT2A cultures, how would this impact the author's conclusions? Also given that loss of ATRX appears to slow the growth of these cells, did the tumors that emerged from intracranially implanted ATRX KO CT2A cells still exhibit loss of ATRX expression?

Third I think it would be useful to acknowledge the limitations of the systems used. Specifically the cells used in this study are tumor cells that evolved in the presence of ATRX and the absence of mutant IDH1 and as such are not driven by either event. Adding both events to these cells may indeed cause changes, but it's not clear that such changes would actually occur in tumors in which loss of ATRX and expression of mutant IDH1 occurred as part of the tumorigenic process. One might delete ATRX in GBM cells and see many changes, none of which would be relevant to how actual ATRX+ GBM cells arose or behaved. Although it may not be possible to avoid this type of limitation, it should be acknowledged.

Finally the language in this submission is a bit loose in suggesting that ATRX loss and mutant IDH1 expression impact the global innate immune response. The innate immune response is quite broad and indeed ATRX loss has previously been shown to suppress the innate immune response driven by dsDNA (it's presumably why ATRX-, mutant IDH1 cells using the ALT mechanism tolerate c-circle DNA). The narrow focus here on the innate immune response driven by dsRNA should therefore not be taken (or implied) to be a general response to all innate immune response activators. Along these lines do the cells here mount an innate immune response to respond to dsDNA and if not might the effects of ATRX loss and IDH1 mutation be much more complex than suggested here?

Reviewer #2:

Remarks to the Author:

This manuscript investigates the observation of the frequent co-occurrence of ATRX and IDH1 mutations in astrocytoma. In brief, the main findings suggest that ATRX deficiency leads to increased innate immune signalling and cytokine secretion in astrocytoma-related immune cells, relative to oligodendroglioma (ATRX-WT cells). Furthermore, IDH1 mutations are suggested to mask these pro-inflammatory effects, and pharmaceutical inhibition of mutant IDH1 relieves such immune suppression. Considering the low responsiveness of diffuse astrocytoma to immunotherapy, these findings possibly indicate a new therapeutic approach in the field. Overall, the findings are interesting, the data are clearly presented and support the claims of the authors. Some comments/questions below:

1. Fig. 1a: a portion of IDH-mutant, 1p/19q non-codeleted ATRX WT astrocytoma also exhibit enrichment for immune GO terms. To better appreciate the difference between ATRX WT and mutant astrocytoma, it would be helpful if the authors could say how many out of the 52 tumours analysed were highly enriched in immune-related pathway, vis a vis the frequency of enrichment in ATRX mutants.

2. I believe that IDH1 and TP53 mutations are both found in nearly 100% of diffuse astrocytoma. Thus, could the authors explain their choice of the 2 diffuse glioma models used:

- a) CT2A, a glioma cell line, which is PTEN deficient and p53 WT
- b) M059J, a glioblastoma cell line deficient for DNA-PK

Do they think that the use of a p53 mutant model would change the growth and/or immune outcomes that they observed?

3. In Figure 1 of Hu et al. (reference 24 in this paper), ATRX loss promoted growth, and this correlated with enhanced DNA repair in IDH1R132H/p53mut astrocytoma cells. Conversely, could the impaired growth observed in ATRX deficient CT2A and M059J cells (Fig. 1e-f) be explained by enhanced DNA damage and cell cycle blockage, as suggested in Fig. 2c? Did the authors look at genome instability in their model(s)?

4. The authors suggest that the immune microenvironment is responsible for limiting tumour growth in ATRX-deficient CT2A glioma. They also show higher T cell and less macrophage recruitment (Fig. 3c and 4a). Nevertheless, CCL2 is a monocyte chemotactic protein that promotes macrophage infiltration – so, how is the increase of CCL2 (Fig. 4b-c) to be reconciled with less macrophage infiltration/recruitment? What about IL-33 levels (where IL-33 is known to protect cancer cells by inhibiting T cell-mediated cancer killing)? Do M059J or RCAS/Ntv-a ATRX-KO tumour-bearing brains also show increased T cell infiltration? What T cells subsets/types are being recruited?

5. Can the authors discuss the reasons for their focus solely on RNA sensing pathways? I understand that this might be linked to the elevated Rigi expression they show (Figure 3), but they also detected elevated Myd88 expression, an adaptor of many TLRs which play an important role in innate immunity as well. More generally, the molecules noted in Figure 3 are themselves ISGs. And what about the DNA sensing pathway through cGAS-STING – where three independent groups (references below) have reported that a loss of ATRX impairs cGAS sensing of dsDNA, and subsequent type I interferon induction?

Chen, Y.A. et al. Extrachromosomal telomere repeat DNA is linked to ALT development via cGAS-STING DNA sensing pathway. *Nat Struct Mol Biol* 24, 1124-1131 (2017).

Floyd, W. et al. Atrx deletion impairs cGAS-STING signaling and increases response to radiation and oncolytic herpesvirus in sarcoma. *bioRxiv*, 2021.03.08.434225 (2021).

Stilp, A.C. et al. The chromatin remodeling protein ATRX positively regulates IRF3-dependent type I interferon production and interferon-induced gene expression. *PLoS Pathog* 18, e1010748 (2022).

6. Inhibition of IDH1R132H did not restore the pro-inflammatory chemokine and cytokine elevation observed in ATRX KO CT2A cells (Fig. 8c). Does this partial rescue suggest that in-vivo inhibition of IDH1R132H would not be sufficient to drive recruitment of immune cells to hinder the growth of ATRX mutant - IDH1 mutant astrocytoma?

Minor points:

i. In Fig. 5a-c, statistics are only provided for 5b. The number of experiments performed for CT2A and RCAS/Ntv-a lines should be indicated, as well as the corresponding statistical analysis.

ii. Lines 265-266: "Idh1R132H cells failed to significantly increase cytokine secretion, though CCL5 and CXCL10 levels were non-significantly elevated" – but CCL5 is starred as significant in Fig. 8c.

iii. In general, the authors should provide WB quantifications and/or state how many time their blots were repeated.

iv. Any hypothesis as to why there is an up-regulation of innate pathways in ATRX KO cells?

v. Fig. 9a: Did the authors look at total STING and pSTING levels by WB? Poly(I:C) stimulation did not induce pIRF3 and pSTAT1 in the control (ATRX WT IDH1 WT, first 2 lanes), while it induced ISG15 and STAT1. Is this something authors observed repeatedly? Both phosphorylation might be expected to occur after Poly(I:C) stimulation, is it then rather a matter of a "missed window"?

vi. Can the authors discuss the (dis)similarities between mouse and human astrocytoma models – noting that different chemokine are plotted in Fig.5a and Fig.5b).

Reviewer #3:

Remarks to the Author:

The manuscript by Hariharan et al. investigates the role for ATRX inactivation in IDH-mutant astrocytoma. Using murine models for glioma, a human glioma line, and data from human gliomas, they investigate the impact of ATRX inactivation on expression of immune response-related genes, secretion of cytokines, and response to immunomodulatory therapy using an immune-competent murine model. While the potential immune manifestations of ATRX inactivation and IDH mutation have been the focus of other studies, the manuscript provides important novel information related to protein level alterations in cytokine secretion, pathway alterations, and the role for ATRX alteration on response to a dsRNA agonist. In addition, the investigators use two different murine models for glioma to support their conclusions.

1. Several of the cytokines with increased secretion in the context of ATRX deletion are considered potent chemokines for monocyte migration and polarization. Given the prominent TAM presence and phenotype in human IDH-mutant astrocytoma, the data in the supplementary files, and some of the prior publications, the data in Figure 4 are surprising. In the CT2A model, macrophages (specific markers used in panel are not specified) in the ATRX ko mice were decreased relative to in control tumors. How do the authors explain this? Were there differences in macrophage function or phenotype with or without ATRX inactivation? In the RCAS/n-tva model how did immune cell subsets compare with and without ATRX inactivation? Did addition of the IDH mutation alter the myeloid phenotype as a function of ATRX inactivation?

2. In Figure 3 the investigators use the nude mouse model as evidence that the dramatic prolongation in OS is associated with microenvironmental factors. While the data from nude mice is suggestive, these mice have many abnormalities in their innate immune response that could complicate this experiment. Antibody depletion of immune cell subsets would more directly test the function of specific immune cell subsets in the phenotype.

3. The data presented in the supplementary figures are very helpful. In the CT2A line, it appears that in the context of IDH mutation ATRX inactivation has an opposite effect on secretion of CCL2, CCL5, and CXCL10 as compared to ATRX inactivation alone (Fig 4b vs. Supp Fig 12b-c). As some of the potential upstream alterations are the same with and without IDH mutation how do the authors explain this? Does data from the RCAS/n-tva model help to clarify this?

Dear Dr. Clancy, Reviewers and Nature Communications editorial team,

We thank you for carefully reviewing our manuscript and providing thoughtful suggestions and insight to improve the quality of our manuscript. We have thoroughly addressed the reviewers' concerns by performing additional experiments and analyses as per their suggestions and incorporated additional information supporting our findings. Point-by-point responses (in black) to the reviewers' comments (in blue) are listed below:

Reviewer #1 - Glioma, IDH/ATRX signaling, preclinical models (Remarks to the Author):

The manuscript "Interplay between ATRX and IDH1..." submitted by Hariharan et al describes studies addressing the role that ATRX loss and IDH1 mutation alone and in combination play in regulating the innate immune response. The key findings of the study are that loss of ATRX expression leads to increased innate immune signaling and cytokine secretion in response to dsRNA stimuli, which in turn appears to be masked by co-expression of mutant IDH1 as is noted in lower-grade astrocytoma. The key take home message of this study is that pharmacologic suppression of IDH1 may in turn reveal the effects of ATRX loss and in turn increase therapeutic response to dsRNA-based therapies. The latter finding is novel, significant, and noteworthy.

There are a number of issues that if addressed would improve the credibility of this work.

- 1. First, technically, it would be very useful to include protein size markers and quantitation in all Western blots displayed. Size markers would help validate the antibodies used while quantitation is required to support claims of differences in expression, especially in cases such as in Fig 7C and 8b where the suggested changes are not immediately apparent.**

We agree with the reviewer's suggestions. Protein size markers have been added to all the Western blot images and corresponding quantitation data has been included in supplementary data. We decided to include quantitation data only for blots that show modest changes in protein expression as including quantitation for every single Western image shown in the manuscript would render the supplementary figure file impractically cumbersome.

- 2. Second there are some technical issues with the cells used in this study. A key finding here is that loss of ATRX expression significantly slows the intracranial growth of cells but only in immunocompetent animals. This is clearly shown in both CT2A and RCAS models. A point glossed over here, however, is that loss of ATRX in CT2A cells also greatly slows the growth of these cells in vitro (Fig 1e), although this apparent >50% decrease is described as "modest" on page 6 and "negligible" in the discussion on page 14.**

We apologize if the wording describing cell growth in our models is not appropriate. We would like to clarify that the data shown in figure 1e refers to the impact of ATRX loss on cell growth, while data shown in figure 6c (now figure 7c) and described in the discussion on page 14 (now page 16) refers to the impact of the presence of IDH1^{R132H} mutation on cell growth. As per Figure 1e, loss of ATRX in the CT2A model leads to a modest decrease in growth, which is not statistically significant (hence the reason for describing the effects on growth as modest). While the *Atrx* KO cells still exhibit delayed growth compared to *Atrx* WT cells (consistent with Figure 1e) when co-expressing IDH1^{R132H} in the CT2A model, this co-expression does not affect growth any further in either background. We have described this effect of IDH1^{R132H} on growth as negligible in the discussion on page 14 (now page 16).

i. What was the exact doubling time of these cultures and is it significantly different in these groups? If it is significantly longer in the ATRX KO CT2A cultures, how would this impact the author's conclusions?

The doubling time of the various lines are as follows: CT2A *Atrx* WT (CRISPR control) line ~27.5 hours; CT2A *Atrx* KO-A ~ 32.23 hours; CT2A *Atrx* KO-B ~ 41.3hours. Consistent with the growth assay data shown in Fig. 1e, *Atrx* KO cells exhibit a relatively longer doubling time, but the difference between *Atrx* WT and *Atrx* KO lines was not statistically significant (P values from Student's t-test: *Atrx* WT compared to *Atrx* KO-A: 0.288; *Atrx* WT compared to *Atrx* KO-B: 0.187). Overall, the modest decrease in growth correlates with the longer doubling times noted in the CT2A *Atrx* KO lines, supporting our original conclusion that ATRX loss has a modest effect on cell growth.

ii. Also given that loss of ATRX appears to slow the growth of these cells, did the tumors that emerged from intracranially implanted ATRX KO CT2A cells still exhibit loss of ATRX expression?

The reviewer has a valid question on whether ATRX loss was maintained in the tumors. Immunohistochemical analysis of both CT2A and RCAS/*Ntv-a* end stage tumors reveal that intracranially implanted ATRX KO cells do maintain loss of ATRX expression. This data has been included as supplementary figure 3d and 3e and text describing this data has been added to the "results" section (lines 159 to 163) as follows:

"Loss of ATRX expression was validated in both CT2A and RCAS/ Ntv-a end-stage tumors (Supplementary Fig. 3d-e). In both cases, Atrx KO glioma cells demonstrated depleted nuclear ATRX labeling by immunohistochemistry, with retained expression in non-neoplastic cellular constituents. By contrast, ATRX-intact control tumors exhibited uniformly strong ATRX expression in tumor cell nuclei."

3. Third I think it would be useful to acknowledge the limitations of the systems used. Specifically the cells used in this study are tumor cells that evolved in the presence of ATRX and the absence of mutant IDH1 and as such are not driven by either event. Adding both events to these cells may indeed cause changes, but its not clear that such changes would actually occur in tumors in which loss of ATRX and expression of mutant IDH1 occurred as part of the tumorigenic process. One might delete ATRX in GBM cells and see many changes, none of which would be relevant to how actual ATRX+ GBM cells arose or behaved. Although it may not be possible to avoid this type of limitation, it should be acknowledged.

We thank the reviewer for this insightful comment. Indeed, the occurrence of the mutations during the tumorigenic process is relevant, though experimentally difficult to implement with our model systems. We have included the following text to the discussion (lines 382 to 414) to address the concurrent onset of mutations in our model as a limitation as well as the importance of the timing and sequential order of the mutations during the tumorigenic process:

"It should be noted that most studies examining the combined pathogenic impact of IDH mutation and ATRX inactivation thus far have addressed these molecular alterations concurrently^{23,24}, whereas in the context of human disease, IDH mutation is thought to arise first followed by the sequential acquisition of ATRX deficiency^{49,50}. With both IDH1 and ATRX abnormalities each exerting independent epigenetic effects on cellular physiology, it is possible that their sequential emergence and interval "priming" of chromatin and transcriptional landscapes could alter ultimate phenotypic expression, accounting for discrepancies in the literature. In this way, the precise timing and order of mutational acquisition could significantly impact therapeutically tractable sensitivities conferred by ATRX deficiency in glioma cells. For instance, one study investigating the effects of ATRX loss in the context of IDH and TP53 mutations concluded that ATRX loss confers an immunosuppressive state as evidenced by upregulated secretion of relevant cytokines and chemokines (CXCL8, IL6, CXCL3, and CSF2) and increased tumor growth²⁴. It is worth noting that the in vivo

arm of that study relied on GL261, a model known for high baseline immunogenicity⁵¹, whereas both our CRISPR and RCAS/Ntv-a models better reflect the low baseline immunity seen in the human context. Another study that developed murine glioma models based on ATRX KO followed by mutant IDH1 overexpression described upregulated pro-inflammatory gene programs and Nfkb1 signaling as characterized by single cell RNAseq and ATAC-seq²³. However, the observed accumulation in this model of immunosuppressive M2 macrophages complicated final conclusions regarding the impact of ATRX loss.

Several strengths and weaknesses are noted in our models. The CT2A glioma line extensively leveraged in this work may already harbor intrinsic tumor-related molecular mechanisms independent of ATRX loss and mutant IDH1 that impact immune microenvironmental engagement. That being said, our RCAS/Ntv-a Atrx KO model, though lacking mutant IDH1, closely mirrors the natural progression of low-grade gliomas in humans and corroborates the immunological effects of ATRX loss noted in human gliomas and the CT2A Atrx KO model. While the aforementioned studies examining the immunologic impact of ATRX deficiency relied heavily on RNA-level data, our work integrated protein and gene set level data, along with functional changes in vivo. Importantly, this study adds to prior work demonstrating the hypersensitivity of ATRX deficient cells to dsRNA-based innate immune agonism. These findings are further supported by our analysis of human TCGA data, showing that combined ATRX loss and IDH mutation in astrocytomas yield a pro-inflammatory phenotype relative to IDH mutation alone."

- 4. Finally the language in this submission is a bit loose in suggesting that ATRX loss and mutant IDH1 expression impact the global innate immune response. The innate immune response is quite broad and indeed ATRX loss has previously been shown to suppress the innate immune response driven by dsDNA (it's presumably why ATRX-, mutant IDH1 cells using the ALT mechanism tolerate c-circle DNA). The narrow focus here on the innate immune response driven by dsRNA should therefore not be taken (or implied) to be a general response to all innate immune response activators. Along these lines do the cells here mount an innate immune response to respond to dsDNA and if not might the effects of ATRX loss and IDH1 mutation be much more complex than suggested here?**

We appreciate the reviewer's concern over our generalized tone that suggested that ATRX loss and IDH mutations impact a global innate immune response. The innate response of ATRX-deficient, ALT-positive cells to dsDNA has already been established by previous studies^{1,2}. Since the impact of ATRX loss on dsDNA-triggered innate signaling has already been examined and our preliminary analysis of ATRX-deficient cells indicated an increased expression of dsRNA sensors, RIG-I and MDA5, we chose to focus on the response of ATRX-deficient glioma cells to dsRNA-based innate agonists. We have now modified the language throughout the manuscript to clarify that the innate response associated with loss of ATRX is dsRNA-driven. We have also discussed previous publications on the impact of ATRX loss on dsDNA-mediated signaling and included the rationale and justification for investigating dsRNA in the discussion section.

So far, our investigations in our ATRX KO models have focused only on dsRNA-dependent innate signaling. We have recently initiated studies into the response of ATRX-deficient cells to dsDNA and other cGAS/STING agonists to better understand the immune responsiveness of ATRX deficient tumors to various innate agonists. Our preliminary data indicates that the innate response to dsDNA is far more complex than what we expected, and we are performing additional experiments to better clarify underlying molecular mechanisms. Hence, we would like to compile the data from these investigations in a separate publication.

We have added the following text to the discussion to address this concern and also modified the text accordingly (lines 335 to 352):

“ATRX loss has been shown to contribute to immune-modulatory behavior through its association with the ALT phenotype and subsequent production of extrachromosomal telomere repeats (ECTR). When ATRX is intact, any extracellular DNA activates the cGAS-STING pathway^{40,41}, triggering a type 1 interferon response with production of IFN β , inhibiting cell growth, and/or promoting cellular elimination. However, recent work has shown that ATRX-mutant and ALT-positive cells may confer defective cytosolic DNA sensing by inactivating the cGAS-STING pathway, precluding the type 1 interferon response^{40,42}. Similar to the downstream effects of the cGAS-STING pathway, activation of dsRNA sensors such as RIG-I and MDA5 ultimately leads to production of type I interferons and other pro-inflammatory cytokines⁴³. We observed an induction in RIG-I and MDA5 in both our human and mouse ATRX-deficient glioma models, indicative of enhanced sensitivity to dsRNA. The precise molecular mechanisms underlying this “primed” cellular state remain unclear. ATRX is a well-established epigenetic regulator and as such, its loss could directly impact the expression of relevant immune pathway genes in the appropriate developmental context, poisoning affected cells for integrated transcriptional responses to dsRNA. ATRX inactivation has also been repeatedly linked to derepression of endogenous retroviral (ERV) elements^{44,45}, which could lead to increased cytoplasmic dsRNA, engagement of antiviral sensing pathways, and interferon responses⁴⁶⁻⁴⁸.”

Reviewer #2 - Nucleic-acid sensing, brain cancer (Remarks to the Author):

This manuscript investigates the observation of the frequent co-occurrence of ATRX and IDH1 mutations in astrocytoma. In brief, the main findings suggest that ATRX deficiency leads to increased innate immune signalling and cytokine secretion in astrocytoma-related immune cells, relative to oligodendroglioma (ATRX-WT cells). Furthermore, IDH1 mutations are suggested to mask these pro-inflammatory effects, and pharmaceutical inhibition of mutant IDH1 relieves such immune suppression. Considering the low responsiveness of diffuse astrocytoma to immunotherapy, these findings possibly indicate a new therapeutic approach in the field. Overall, the findings are interesting, the data are clearly presented and support the claims of the authors. Some comments/questions below:

- 1. Fig. 1a: a portion of IDH-mutant, 1p/19q non-codeleted ATRX WT astrocytoma also exhibit enrichment for immune GO terms. To better appreciate the difference between ATRX WT and mutant astrocytoma, it would be helpful if the authors could say how many out of the 52 tumours analysed were highly enriched in immune-related pathway, vis a vis the frequency of enrichment in ATRX mutants.**

We have added the following text to the figure legend for Fig 1a, indicating the number of ATRX WT and ATRX mutant astrocytomas that show positive enrichment for the various immune related pathways:

“About 55% (28 out of 52) of 1p-19q noncodelet/ IDH mutant/ ATRX WT astrocytomas and 62% (118 out of 191) of 1p-19q noncodelet/ IDH mutant/ ATRX mutant astrocytomas exhibited positive enrichment for these pathways.”

We have also included a table listing frequency of positive enrichment (Supplementary table 1) to support our data. It should be noted that, regardless of ATRX mutational status, the vast majority of IDH-mutant astrocytomas express very low levels of ATRX, with alternative mechanisms of transcriptional silencing likely involved³. Accordingly, we are not surprised at the similar degree to which nominally ATRX-mutant and ATRX-wildtype IDH-mutant astrocytomas exhibit positive immune pathway enrichment.

- 2. I believe that IDH1 and TP53 mutations are both found in nearly 100% of diffuse astrocytoma. Thus, could the authors explain their choice of the 2 diffuse glioma models used:**
 - a) CT2A, a glioma cell line, which is PTEN deficient and p53 WT**
 - b) M059J, a glioblastoma cell line deficient for DNA-PK**

We agree with the reviewer that *IDH1* and *TP53* mutations are found in a majority of diffuse astrocytoma patients. Our primary goal was to first delineate the impact of ATRX loss (in the presence of WT *IDH1*) on immune signaling in general and innate immune signaling in particular. To ask these questions, we required a cell line model with intact innate signaling pathways. While some glioma cell lines possess genomic alterations to immune relevant gene loci, such as deletion of interferon loci, we chose CT2A and M059J to generate our ATRX intact/ ATRX-deficient isogenic models as they are well established glioma models with intact innate immune signaling pathways. We have now included this information in the results section (lines 135 to 137) as follows:

“To investigate the role of ATRX deficiency in regulating inflammation in gliomas, we developed both mouse and human experimental systems using glioma cell lines that exhibit intact innate immune signaling pathways.”

3. Do they think that the use of a p53 mutant model would change the growth and/or immune outcomes that they observed?

The RCAS/Ntv-a model used in this study is a p53-deficient model, while CT2A is p53-intact. The immune outcomes that we have observed, including enrichment for immune-based signaling, increased baseline inflammation and cytokine secretion, and increased sensitivity to dsRNA agonists were similar in both the p53-deficient RCAS/Ntv-a model and p53-intact CT2A model, indicating that based on our current data, p53 status does not appear to affect tumor growth or immune signaling.

4. In Figure 1 of Hu et al. (reference 24 in this paper), ATRX loss promoted growth, and this correlated with enhanced DNA repair in *IDH1R132H/p53mut* astrocytoma cells. Conversely, could the impaired growth observed in ATRX deficient CT2A and M059J cells (Fig. 1e-f) be explained by enhanced DNA damage and cell cycle blockage, as suggested in Fig. 2c? Did the authors look at genome instability in their model(s)?

We agree with the reviewer’s reasoning that enhanced DNA damage and cell cycle blockage might account for an element of impaired growth in our ATRX-deficient cell line models. Indeed, previous publications have defined a role for ATRX in DNA repair and damage responses⁴. Moreover, loss of ATRX is also associated with dysregulated cell cycle transition⁵. However, in our *in vivo* CT2A xenograft studies (Fig. 3c), we found that tumor growth in an immune-deficient context completely abrogated differences in growth kinetics between ATRX-deficient and ATRX-intact isogenics. Accordingly, we reasoned that microenvironmental distinctions were more relevant to this specific phenotype and focused the paper on their further mechanistic elucidation. We have yet to explore genomic instability and its impact of *in vivo* growth of our cell line models and intend to do so in future studies.

5. The authors suggest that the immune microenvironment is responsible for limiting tumour growth in ATRX-deficient CT2A glioma. They also show higher T cell and less macrophage recruitment (Fig. 3c and 4a).

i. Nevertheless, CCL2 is a monocyte chemotactic protein that promotes macrophage infiltration – so, how is the increase of CCL2 (Fig. 4b-c) to be reconciled with less macrophage infiltration/recruitment?

First, we apologize for any confusion caused by the text describing Fig. 4a; we have now revised our description of this figure to make clear that these data are from tumor bearing hemispheres (not dissected tumors) analyzed at the same time point after tumor implantation. At this time point, tumors in CT2A CRISPR control (*Atrx* WT) implanted mice were macroscopically evident/end stage, whereas tumors in CT2A *Atrx* KO- A or KO-B implanted mice were not (consistent with Fig. 3c). To directly address the immune cell content in the immediate tumor microenvironment, as opposed to the entire brain hemisphere, we have now included immunohistochemical data (New Fig 4b and 4c). By IHC, we observe

increased F4/80 staining in *Atrx* KO tumors relative to CRISPR controls, consistent with higher macrophage influx within the tumor itself. For Fig. 4a, given that the % live cell metric encompasses both normal brain (e.g. microglia) and that the CT2A model naturally recruits high densities of macrophages with tumor progression, we suspect that the lower macrophage number in mice bearing *Atrx* KO xenografts largely reflects their more modest tumor burden. Consistent with this explanation, histological analysis of *Atrx* KO tumor microenvironment (Fig 4b)—as opposed to tumor bearing hemispheres—revealed markedly increased F4/80 staining. We have clarified these differences in the revised manuscript (lines 192 to 208), pasted below for convenience:

“To this end, we used flow cytometry to analyze dissociated $Atrx^{WT}$ and $Atrx$ -KO CT2A tumor-bearing brain hemispheres from immunocompetent mice 14 days after intracranial implantation (Fig. 4a; Supplementary Fig. 6). At this time point, gliomas were not macroscopically evident in $Atrx$ -KO CT2A tumor-bearing mice, prompting our analysis of the entire xenografted hemisphere for both $Atrx$ -KO and $Atrx^{WT}$ cases. ATR X loss led to a significant increase in CD3+ T cells and, in particular, CD4+ T-cells within tumor bearing hemispheres in allografted mice, while macrophage infiltration was reduced. Trends toward increases in multiple T-cells subsets were also observed in RCAS tumor xenografts by the same experimental paradigm, although differences did not reach formal statistical significance (Supplementary Fig. 8a). By contrast, flow cytometry analysis demonstrated reduced overall macrophage levels in ATR X-deficient xenografts relative to ATR X-intact controls (Fig. 4a). We reasoned that this discrepancy might simply reflect the lower tumor size of ATR X-deficient xenografts, and indeed by immunohistochemistry, we found higher levels of CD45 and F4/80 cell labeling in both $Atrx$ KO-A and KO-B tumors than in CRISPR controls (Fig. 4b; Supplementary Fig. 7a). Once again, these findings were supported by similar trends in the RCAS model, and CD3 IHC results approached statistical significance ($p=0.0649$) (Fig. 4b, c; Supplementary Fig. 8b).”

ii. What about IL-33 levels (where IL-33 is known to protect cancer cells by inhibiting T cell-mediated cancer killing)?

We queried IL-33 in our RNA-sequencing data sets and found that while IL33 was downregulated in CT2A *Atrx* KO-B cell lines compared to CRISPR control (*Atrx* WT), it was not significantly altered in the CT2A *Atrx* KO-A line. IL33 was also downregulated in RCAS/ *Ntv-a Atrx* KO cell lines and tumors. On the other hand, IL33 was upregulated in the M059J *ATR*X KO line compared to the corresponding *ATR*X WT line. So based on our preliminary gene expression analysis of IL33 in mouse and human cell lines and tumors, there is no clear trend as to how *ATR*X loss may influence IL33 levels in our models.

iii. Do M059J or RCAS/*Ntv-a ATR*X-KO tumour-bearing brains also show increased T cell infiltration?

Since M059J is a human glioma cell line and cannot be implanted in C57BL/6 immunocompetent mice, we have not been able to determine if *ATR*X KO is also associated with increased T-cell infiltration in this model. However, we now include flow cytometry-based studies on RCAS/*nTva* tumors. While our findings do not reach formal statistical significance, we do observe trends toward increased T cell and overall immune cell infiltration in *ATR*X-deficient tumors relative to *ATR*X-intact counterparts. These findings are further supported by IHC for CD3 and CD45, which show similar trends toward increased tumor infiltration in the *ATR*X-deficient context, with CD3 labeling approaching statistical significance. Taken together, these findings concur with our previously shown flow data in the CT2A model (Fig. 4a) and strengthen our conclusions regarding immune cell infiltration. These data are presented in supplementary fig. 8 and described on pages 8 and 9 (lines 199 to 208).

“Trends toward increases in multiple T-cells subsets were also observed in RCAS tumor xenografts by the same experimental paradigm, although differences did not reach formal statistical significance

(Supplementary Fig. 8a). By contrast, flow cytometry analysis demonstrated reduced overall macrophage levels in *ATRX*-deficient xenografts relative to *ATRX*-intact controls (Fig. 4a). We reasoned that this discrepancy might simply reflect the lower tumor size of *ATRX*-deficient xenografts, and indeed by immunohistochemistry, we found higher levels of CD45 and F4/80 cell labeling in both *Atrx* KO-A and KO-B tumors than in CRISPR controls (Fig. 4b; Supplementary Fig. 7a). Once again, these findings were supported by similar trends in the RCAS model, and CD3 IHC results approached statistical significance ($p=0.0649$) (Fig. 4b, c; Supplementary Fig. 8b).”

iv. What T cells subsets/types are being recruited?

Because tumor bearing hemispheres were analyzed as opposed to dissected tumor in Figure 4a, and tumor burden was substantially lower in *Atrx* KO tumors, we were concerned that differences in T cell subsets/phenotypes may have been influenced by the higher tumor burdens seen in mice harboring *ATRX*-intact xenografts. Indeed, we have observed that end stage CT2A tumors exhibit high levels of total T cells, but that many exhibit an exhausted phenotype. We believe that the influx of higher T cell numbers in *Atrx* KO tumors is related to loss of *ATRX* and not differences in tumor burden, as there were increased T cell numbers within *Atrx* KO tumors relative to the numbers of macrophages (Fig. 4a), T cell signatures were observed in RNA-sequencing data (Fig. 2c), and tumor growth differences by *ATRX* status were minimal in nude mice lacking T cells (Fig. 3c). Below we show T cell subtype and phenotype data from one of two repeat experiments conducted for Fig 4a. We found decreased exhaustion markers (PD1 and TIM3) for both CD4 and CD8 T cells in *Atrx*-KO tumors relative to controls, while levels of CD69+ (activation) and CD62L^{neg} (effector) cells were comparatively unchanged. Notably, these analyses are from one of the two repeats, and TIM3 and PD1 expression are lower in normal brain, which caused our hesitation in including these data as they may merely reflect lacking tumor burden and not direct effects of *ATRX* KO.

6. Can the authors discuss the reasons for their focus solely on RNA sensing pathways? I understand that this might be linked to the elevated *Rigi* expression they show (Figure 3), but they also detected elevated *Myd88* expression, an adaptor of many TLRs which play an important role in innate immunity as well. More generally, the molecules noted in Figure 3 are themselves ISGs. And what about the DNA sensing pathway through cGAS-STING – where three independent groups (references below) have reported that a loss of *ATRX* impairs cGAS sensing of dsDNA, and subsequent type I interferon induction?

- ◇ Chen, Y.A. et al. Extrachromosomal telomere repeat DNA is linked to ALT development via cGAS-STING DNA sensing pathway. Nat Struct Mol Biol 24, 1124-1131 (2017).
- ◇ Floyd, W. et al. Atrx deletion impairs cGAS-STING signaling and increases response to radiation and oncolytic herpesvirus in sarcoma. bioRxiv, 2021.03.08.434225 (2021).
- ◇ Stilp, A.C. et al. The chromatin remodeling protein ATRX positively regulates IRF3- dependent type I interferon production and interferon-induced gene expression. PLoS Pathog 18, e1010748 (2022).

We agree with the reviewer that previous studies have already reported the impaired innate response of ATRX-deficient cancer cells to dsDNA. The impact of ATRX loss on dsDNA-triggered innate signaling has already been established. Our preliminary analysis of ATRX-deficient cells indicated an increased expression of dsRNA sensors, RIG-I and MDA5 and this has not previously been reported so is novel. Thus, we chose to focus on the response of ATRX-deficient glioma cells to dsRNA-based innate agonists. Data shown in new figures 6a – 6e suggest that in addition to the elevated expression of RIG-I and MDA5, loss in ATRX is associated with increased responsiveness to polyIC, a dsRNA agonist, which we believe may be an inherent vulnerability of ATRX-deficient gliomas with the potential for therapeutic exploitation. Despite the differences in response to dsDNA shown by other groups and response to dsRNA shown by our work, we observe that ATRX loss is associated with an overall pro-inflammatory phenotype as supported by RNAseq and protein expression analysis across three different model systems. We now include the text below (in italics) in the discussion section and also modify the existing text (lines 335 to 352), wherein we discuss previous reports on the impact of ATRX loss on dsDNA-mediated signaling and clarify the rationale and justification for investigating dsRNA.

So far, our investigations in ATRX KO models have focused only on dsRNA-dependent innate signaling. We have recently initiated studies into the response of ATRX-deficient cells to dsDNA and other cGAS/ STING agonists to gain a better understanding of the immune responsiveness of ATRX deficient tumors to various innate agonists. Our preliminary data indicates that the innate response to dsDNA is far more complex than expected, and we are performing additional experiments to better understand the relevant molecular foundations. Accordingly, we would like to compile these data for a separate publication.

“ ATRX loss has been shown to contribute to immune-modulatory behavior through its association with the ALT phenotype and subsequent production of extrachromosomal telomere repeats (ECTR). When ATRX is intact, any extracellular DNA activates the cGAS-STING pathway^{40,41}, triggering a type 1 interferon response with production of IFN β , inhibiting cell growth, and/or promoting cellular elimination. However, recent work has shown that ATRX-mutant and ALT-positive cells may confer defective cytosolic DNA sensing by inactivating the cGAS-STING pathway, precluding the type 1 interferon response^{40,42}. Similar to the downstream effects of the cGAS-STING pathway, activation of dsRNA sensors such as RIG-I and MDA5 ultimately leads to production of type I interferons and other pro-inflammatory cytokines⁴³. We observed an induction in RIG-I and MDA5 in both our human and mouse ATRX-deficient glioma models, indicative of enhanced sensitivity to dsRNA. The precise molecular mechanisms underlying this “primed” cellular state remain unclear. ATRX is a well-established epigenetic regulator and as such, its loss could directly impact the expression of relevant immune pathway genes in the appropriate developmental context, poisoning affected cells for integrated transcriptional responses to dsRNA. ATRX inactivation has also been repeatedly linked to derepression of endogenous retroviral (ERV) elements^{44,45}, which could lead to increased cytoplasmic dsRNA, engagement of antiviral sensing pathways, and interferon responses⁴⁶⁻⁴⁸.”

7. Inhibition of IDH1R132H did not restore the pro-inflammatory chemokine and cytokine elevation observed in ATRX KO CT2A cells (Fig. 8c). Does this partial rescue suggest that in-vivo inhibition of

IDH1R132H would not be sufficient to drive recruitment of immune cells to hinder the growth of ATRX mutant - IDH1 mutant astrocytoma?

This is an astute observation by the reviewer. It is worth noting that while the restoration of expression of dsRNA sensing genes was apparent with mIDH1 inhibition, there was no obvious rescue of cytokine production in the *in vitro* context. The impact of mIDH inhibition on immune recruitment and tumor growth *in vivo* is an area of active investigation and we hope will be the subject of a future manuscript. Based on our preliminary *in vitro* data, we suspect that a combination of mIDH1 inhibition and dsRNA agonism is necessary to impair tumor growth. Additionally, experiments investigating the impact of duration of mIDH inhibition on reversal of immune suppression may also be warranted.

Minor points:

- 1. In Fig. 5a-c, statistics are only provided for 5b. The number of experiments performed for CT2A and RCAS/Ntv-a lines should be indicated, as well as the corresponding statistical analysis.**

We apologize for not providing statistics for figures 5a and 5c. These figures have now been modified to include statistics, and the number of replicates has been included in the figure legend.

- 2. Lines 265-266: “Idh1R132H cells failed to significantly increase cytokine secretion, though CCL5 and CXCL10 levels were non-significantly elevated” – but CCL5 is starred as significant in Fig. 8c.**

We thank the reviewer for catching this significant point. The text in lines 265-266 (lines 283-284) has now been modified to match with the data shown in figure 8c (new figure 9c):

“Specifically, while CCL5 levels were robustly upregulated, other cytokines, including CCL2 and CXCL10, showed only slight increases that did not reach statistical significance.”

- 3. In general, the authors should provide WB quantifications and/or state how many time their blots were repeated.**

We have now included quantitation data for Western blot images that show modest changes in protein expression. We have also included the number of times these blots were repeated in the corresponding figure legends.

- 4. Any hypothesis as to why there is an up-regulation of innate pathways in ATRX KO cells?**

ATRX is a SWI/SNF family member and a well-established epigenetic regulator that impacts chromatin accessibility and underlying transcription genome-wide, particularly in cells of neuroepithelial lineage⁶. Accordingly, ATRX loss could directly impact the expression of relevant immune pathway genes. ATRX inactivation has also been repeatedly linked to the derepression of repetitive endogenous retroviral regions (ERVs)^{7,8}. Inappropriate ERV expression could presumably lead to increased cytoplasmic dsRNA, activation of antiviral sensing pathways, and interferon responses⁹⁻¹¹. This effect was observed on a larger scale in TCGA analyses where negative correlations between SWI/SNF member expression and ERV transcript levels were observed¹⁰. We have included this hypothesis in the discussion (lines 344 to 352):

“We observed an induction in RIG-I and MDA5 in both our human and mouse ATRX-deficient glioma models, indicative of enhanced sensitivity to dsRNA. The precise molecular mechanisms underlying this “primed” cellular state remain unclear. ATRX is a well-established epigenetic regulator and as such, its loss could directly impact the expression of relevant immune pathway genes in the appropriate developmental context, poisoning affected cells for integrated transcriptional responses to dsRNA. ATRX inactivation has also

been repeatedly linked to derepression of endogenous retroviral (ERV) elements^{44,45}, which could lead to increased cytoplasmic dsRNA, engagement of antiviral sensing pathways, and interferon responses⁴⁶⁻⁴⁸.

5. Fig. 9a: Did the authors look at total STING and pSTING levels by WB?

Poly(I:C) stimulation did not induce pIRF3 and pSTAT1 in the control (ATRX WT IDH1 WT, first 2 lanes), while it induced ISG15 and STAT1. Is this something authors observed repeatedly? Both phosphorylation might be expected to occur after Poly(I:C) stimulation, is it then rather a matter of a “missed window”?

We did not evaluate pSTING and STING levels in Fig. 9a. As indicated above (Reviewer #1 comment 4), we have begun to address the complex mechanisms by which the innate response to dsDNA influences the ATRX-deficient immune microenvironment and are packaging this work in a separate manuscript.

The differential impact of poly (I:C) on pIRF3 and pSTAT1 in ATRX-intact and ATRX-deficient glioma models likely speaks to an underlying mechanism involving shifts in epigenomic landscapes and transcription, rendering ATRX-deficient cells “poised” to respond via their dsRNA machinery. In this context, higher endogenous expression of key machinery elements, like RIG-I or MDA5, would promote enhanced responsiveness in the setting of exogenous poly(I:C). As indicated in the response to the previous comment, ERV derepression could also be involved in sensitizing ATRX-deficient cells to dsRNA stimulation. We anticipate additional mechanistic characterization in future studies.

6. Can the authors discuss the (dis)similarities between mouse and human astrocytoma models – noting that different chemokine are plotted in Fig.5a and Fig.5b)?

We would like to clarify that different cytokines were plotted in figures 5a, 5b and 5c primarily because a commercially available cytokine panel specific to mouse was used for figures 5a and 5c, while cytokine panels specific to human were used for figure 5b. We note that while some cytokines including CCL5, CXCL10, IL6, IFN β , are included in both the mouse and human panels, several other cytokines are unique to one or the other. We have clarified in the results section that mouse and human-specific cytokine panels were used for respective sample sets (Line 220). Since the primary goal of these experiments was to determine if poly (I:C) treatment leads to a heightened secretion of pro-inflammatory cytokines in the ATRX KO cells, we focused on the overall cytokine response rather than the levels of individual cytokines. Finally, as these studies ultimately feature a relatively small panel of cytokines, we are hesitant to draw broad conclusions about (dis)similarities between mouse and human astrocytoma models with regard to cytokine secretion.

Reviewer #3 - Glioma, immunotherapy, preclinical models, scRNA-seq (Remarks to the Author):

The manuscript by Hariharan et al. investigates the role for ATRX inactivation in IDH- mutant astrocytoma. Using murine models for glioma, a human glioma line, and data from human gliomas, they investigate the impact of ATRX inactivation on expression of immune response-related genes, secretion of cytokines, and response to immunomodulatory therapy using an immune-competent murine model. While the potential immune manifestations of ATRX inactivation and IDH mutation have been the focus of other studies, the manuscript provides important novel information related to protein level alterations in cytokine secretion, pathway alterations, and the role for ATRX alteration on response to a dsRNA agonist. In addition, the investigators use two different murine models for glioma to support their conclusions.

1. Several of the cytokines with increased secretion in the context of ATRX deletion are considered potent chemokines for monocyte migration and polarization. Given the prominent TAM presence and phenotype in human IDH-mutant astrocytoma, the data in the supplementary files, and some of the prior publications, the data in Figure 4 are surprising.

i. In the CT2A model, macrophages (specific markers used in panel are not specified) in the ATRX ko mice were decreased relative to in control tumors. How do the authors explain this?

We apologize for not providing the strategy used to identify macrophages in the figure legend. We have now included that information in the legend to Figure 4a. We also apologize for any confusion caused by the text describing Figure 4a; we have now revised our description of this figure to make clear that these data are from tumor bearing hemispheres (not dissected tumors) analyzed at the same time point after tumor implantation. At this time point, tumors in CT2A CRISPR control (*Atrx* WT) implanted mice were macroscopically evident/end stage, whereas tumors in CT2A *Atrx* KO- A or KO-B implanted mice were much smaller (consistent with Figure 3c). To directly address the immune cell content in the immediate tumor microenvironment, as opposed to the entire brain hemisphere, we have now included immunohistochemical data (New Fig 4b and 4c). By IHC, we observe increased F4/80 staining in *Atrx* KO tumors relative to CRISPR controls, consistent with higher macrophage influx within the tumor itself. For Figure 4a, given that the % live cell metric encompasses both normal brain (e.g. microglia) and that the CT2A model naturally recruits high densities of macrophages with tumor progression, we suspect that the lower macrophage number in mice bearing *Atrx* KO xenografts largely reflects their more modest tumor burden. Consistent with this explanation, histological analysis of *Atrx* KO tumor microenvironment (Fig 4b)—as opposed to tumor bearing hemispheres—revealed markedly increased F4/80 staining. We have clarified these differences in the revised manuscript (lines 192 to 208), pasted below for convenience:

“To this end, we used flow cytometry to analyze dissociated Atrx^{WT} and Atrx-KO CT2A tumor-bearing brain hemispheres from immunocompetent mice 14 days after intracranial implantation (Fig. 4a; Supplementary Fig. 6). At this time point, gliomas were not macroscopically evident in Atrx-KO CT2A tumor-bearing mice, prompting our analysis of the entire xenografted hemisphere for both Atrx-KO and Atrx^{WT} cases. ATRX loss led to a significant increase in CD3+ T cells and, in particular, CD4+ T-cells within tumor bearing hemispheres in allografted mice, while macrophage infiltration was reduced. Trends toward increases in multiple T-cells subsets were also observed in RCAS tumor xenografts by the same experimental paradigm, although differences did not reach formal statistical significance (Supplementary Fig. 8a). By contrast, flow cytometry analysis demonstrated reduced overall macrophage levels in ATRX-deficient xenografts relative to ATRX-intact controls (Fig. 4a). We reasoned that this discrepancy might simply reflect the lower tumor size of ATRX-deficient xenografts, and indeed by immunohistochemistry, we found higher levels of CD45 and F4/80 cell labeling in both Atrx KO-A and KO-B tumors than in CRISPR controls (Fig. 4b; Supplementary Fig. 7a). Once again, these findings were supported by similar trends in the RCAS model, and CD3 IHC results approached statistical significance (p=0.0649) (Fig. 4b-c; Supplementary Fig. 8b).”

ii. Were there differences in macrophage function or phenotype with or without ATRX inactivation?

Performance of these analyses are similarly complicated by our strategy analyzing immune cell content in implanted hemispheres at equivalent time points (see response to previous comment above. We did assess macrophage phenotypes in one of the repeat experiments of Figure 4a (See Figure below). As

expected, the macrophage phenotypes seen in *Atrx* KO mice resemble those of normal brain, as a consequence of the more modest tumor burden exhibited by these mice. We chose not to include these data, therefore, as they likely reflect differences in the composition of macrophage subsets/origins rather than distinct activation phenotypes, as a consequence of discrepant tumor burdens in control and ATRX-deficient tumor bearing mice.

iii. In the RCAS/n-tva model how did immune cell subsets compare with and without ATRX inactivation?

We now include flow cytometry-based studies on RCAS/nTva tumors. While our findings do not reach formal statistical significance, we do observe trends toward increased T cell and overall immune cell infiltration in ATRX-deficient tumor relative to ATRX-intact counterparts. These findings are further supported by IHC for CD3 and CD45, which show similar trends toward increased tumor infiltration in the ATRX-deficient context, with CD3 labeling approaching statistical significance. Taken together, these findings concur with our previously shown flow data from the CT2A model (Fig. 4a) and strengthen our conclusions regarding immune cell infiltration. These data are presented in supplementary fig. 8 and described on pages 8 and 9 (lines 199 to 208):

*“Trends toward increases in multiple T-cells subsets were also observed in RCAS tumor xenografts by the same experimental paradigm, although differences did not reach formal statistical significance (Supplementary Fig. 8a). By contrast, flow cytometry analysis demonstrated reduced overall macrophage levels in ATRX-deficient xenografts relative to ATRX-intact controls (Fig. 4a). We reasoned that this discrepancy might simply reflect the lower tumor size of ATRX-deficient xenografts, and indeed by immunohistochemistry, we found higher levels of CD45 and F4/80 cell labeling in both *Atrx* KO-A and KO-B tumors than in CRISPR controls (Fig. 4b; Supplementary Fig. 7a). Once again, these findings were supported by similar trends in the RCAS model, and CD3 IHC results approached statistical significance ($p=0.0649$) (Fig. 4b-c; Supplementary Fig. 8b).”*

iv. Did addition of the IDH mutation alter the myeloid phenotype as a function of ATRX inactivation?

We did not assess IDH-mutated ATRX KO tumors by flow cytometry due to the complexities of interpreting data from mice with divergent tumor burdens (as apparent from our analyses in Figure 4a).

2. In Figure 3 the investigators use the nude mouse model as evidence that the dramatic prolongation in OS is associated with microenvironmental factors. While the data from nude mice is suggestive, these mice have many abnormalities in their innate immune response that could complicate this experiment. Antibody depletion of immune cell subsets would more directly test the function of specific immune cell subsets in the phenotype.

We agree with the reviewer that antibody depletion experiments would more directly assess the role of specific immune cell subsets in the observed microenvironmental phenotype. These challenging experiments are ongoing and will inform future manuscripts. For the purposes of this study, we believe that the nude mice approach sufficiently establishes an immune microenvironmental mechanism driving prolonged survival in ATRX-deficient glioma models, with subsequent figures documenting altered immune cell infiltration and mechanistically implicating hyperagonizable innate immune pathways. Accordingly, our overall approach clarifies underlying biological mechanism despite its employment of nude mice in its initial phenotypic characterizations.

3. The data presented in the supplementary figures are very helpful. In the CT2A line, it appears that in the context of IDH mutation ATRX inactivation has an opposite effect on secretion of CCL2, CCL5, and CXCL10 as compared to ATRX inactivation alone (Fig 4b vs. Supp Fig 12b-c). As some of the potential

**upstream alterations are the same with and without IDH mutation how do the authors explain this?
Does data from the RCAS/n-tva model help to clarify this?**

We would like to take this opportunity to clarify our figures. Data shown in the original figure 4b (new figure 5a) indicate that cytokines like CCL2, CCL5 and CXCL10 are upregulated in ATRX-deficient (IDH1-WT) CT2A cells. On the other hand, data shown in the original figure 8c (new figure 9c) and associated original supplementary figure 12c (new figure 13e) indicate that while inhibition of mutant IDH1 restores cytokine levels in the *Atrx* WT/ mIDH1 line, it does not have a significant effect in further inducing cytokine expression in the *Atrx* KO/ mIDH1 line. Therefore, 1) ATRX loss promotes increased cytokine secretion, 2) co-occurring mutant IDH1 dampens cytokine secretion as shown in new Fig 8d, and 3) cytokine levels are not significantly altered upon mutant IDH1 inhibition. We believe that the immunosuppressive effects of mutant IDH1 are a major contributor to the dampened cytokine secretion seen in mutant IDH1-expressing lines. We agree with the reviewer's observation that expression of upstream signaling mediators like total TBK1 and IRF3 are unchanged in the *Atrx* KO/ IDH1 WT line compared to *Atrx* KO/ IDH1^{R132H} line. However, several other immune modulators including RIG-I, MDA5, STAT1, IRF7 and IRF9 are downregulated in the IDH1^{R132H} expressing line (New fig. 8b-c), which may contribute to weakened cytokine response. We have yet to study the effects of mutant IDH1 in the RCAS/ Ntv-a model, as we have been unsuccessful in our attempts to obtain a sufficient expression level for the mutant protein thus far. We are currently generating knock-in reagents; however, these studies are beyond the scope of the current manuscript.

References:

1. Chen, Y.-A., *et al.* Extrachromosomal telomere repeat DNA is linked to ALT development via cGAS-STING DNA sensing pathway. *Nature Structural & Molecular Biology* **24**, 1124-1131 (2017).
2. Floyd, W., *et al.* *Atrx* deletion impairs CGAS/STING signaling and increases sarcoma response to radiation and oncolytic herpesvirus. *J Clin Invest* **133**(2023).
3. Brat, D.J., *et al.* Comprehensive, Integrative Genomic Analysis of Diffuse Lower-Grade Gliomas. *The New England journal of medicine* **372**, 2481-2498 (2015).
4. Aguilera, P. & Lopez-Contreras, A.J. ATRX, a guardian of chromatin. *Trends Genet* **39**, 505-519 (2023).
5. Qin, T., *et al.* ATRX loss in glioma results in dysregulation of cell-cycle phase transition and ATM inhibitor radio-sensitization. *Cell Rep* **38**, 110216 (2022).
6. Danussi, C., *et al.* *Atrx* inactivation drives disease-defining phenotypes in glioma cells of origin through global epigenomic remodeling. *Nature Communications* **9**, 1057 (2018).
7. Sadic, D., *et al.* *Atrx* promotes heterochromatin formation at retrotransposons. *EMBO Rep* **16**, 836-850 (2015).
8. Valenzuela, M., Amato, R., Sgura, A., Antoccia, A. & Berardinelli, F. The Multiple Facets of ATRX Protein. *Cancers (Basel)* **13**(2021).
9. Chiappinelli, K.B., *et al.* Inhibiting DNA Methylation Causes an Interferon Response in Cancer via dsRNA Including Endogenous Retroviruses. *Cell* **162**, 974-986 (2015).
10. Canadas, I., *et al.* Tumor innate immunity primed by specific interferon-stimulated endogenous retroviruses. *Nat Med* **24**, 1143-1150 (2018).
11. Roulois, D., *et al.* DNA-Demethylating Agents Target Colorectal Cancer Cells by Inducing Viral Mimicry by Endogenous Transcripts. *Cell* **162**, 961-973 (2015).

Reviewers' Comments:

Reviewer #1:

Remarks to the Author:

The manuscript "Interplay between ATRX and IDH1..." submitted by Hariharan et al describes studies addressing the role that ATRX loss and IDH1 mutation alone and in combination play in regulating the innate immune response. The key findings of the study are that loss of ATRX expression leads to increased innate immune signaling and cytokine secretion in response to dsRNA stimuli, which in turn appears to be masked by co-expression of mutant IDH1 as is noted in lower-grade astrocytoma. The key take home message of this study is that pharmacologic suppression of IDH1 may in turn reveal the effects of ATRX loss and in turn increase therapeutic response to dsRNA-based therapies. The latter finding is novel, significant, and noteworthy. The revised version of this manuscript does a good job addressing the extensive points raised by the reviewers and is significantly improved relative to the initial submission.

Reviewer #2:

None

Reviewer #3:

Remarks to the Author:

The authors have improved the clarity of the manuscript and responded to this Reviewer's comments.